**Stable isotopic evidence for the excess leaching of unprocessed atmospheric**
**nitrate from forested catchments under high nitrogen saturation**

Weitian Ding[1], Urumu Tsunogai[1], Fumiko Nakagawa[1], Takashi Sambuichi[1], Masaaki

Chiwa[2], Tamao Kasahara[3], Ken'ichi Shinozuka[4]

[1]Graduate School of Environmental Studies, Nagoya University, Furo-cho, Chikusa-ku,

Nagoya 464-8601, Japan

[2]Kyushu University Forest, Kyushu University, Japan

[3]Faculty of Agriculture, Kyushu University, Japan

[4]River Basin Research Center, Gifu University, 1-1 Yanagido, Gifu, 501-1193, Japan

Corresponding author: Weitian Ding

Email: ding.weitian.v2@s.mail.nagoya-u.ac.jp

**Abstract**
Owing to the elevated loading of nitrogen through atmospheric deposition, some
forested ecosystems become nitrogen saturated, from which elevated levels of nitrate
are exported. The average concentration of stream nitrate eluted from upstream and
downstream of the Kasuya Research forested catchments (FK1 and FK2 catchments)
in Japan were more than 90 μM, implying that these forested catchments were under
nitrogen saturation. To verify that these forested catchments were under the nitrogen
saturation, we determined the export flux of unprocessed atmospheric nitrate relative to
the entire deposition flux ($M_{atm}/D_{atm}$ ratio) in these catchments, because the $M_{atm}/D_{atm}$
ratio has recently been proposed as a reliable index to evaluate nitrogen saturation in
forested catchments. Specifically, we determined the temporal variation in the
concentrations and stable isotopic compositions, including $\Delta^{17}O$, of stream nitrate in
the FK catchments for more than 2 years. In addition, for comparison, the same
parameters were also monitored in the Shiiba Research forested catchment (MY
catchment) in Japan during the same period, where the average stream nitrate
concentration was low, less than 10 μM. While showing the average nitrate
concentrations of 109.5, 90.9, and 7.3 μM in FK1, FK2, and MY, respectively, the
catchments showed average $\Delta^{17}O$ values of +2.6, +1.5, and +0.6 ‰ in FK1, FK2, and
MY, respectively. Thus, the average concentration of unprocessed atmospheric nitrate
($[NO_3^-{}_{atm}]$) was estimated to be 10.8, 5.1, and 0.2 μM in FK1, FK2, and MY,
respectively, and the $M_{atm}/D_{atm}$ ratio was estimated to be 14.1, 6.6, and 1.3 % in FK1,
FK2, and MY, respectively. The estimated $M_{atm}/D_{atm}$ ratio in FK1 (14.1 %) was the
highest ever reported from temperate forested catchments monitored for more than 1
year. Thus, we concluded that nitrogen saturation was responsible for the enrichment
of stream nitrate in the FK catchments, together with the elevated $NO_3^-{}_{atm}$ leaching
from the catchments. While the stream nitrate concentration ($[NO_3^-]$) can be affected
by the amount of precipitation, the $M_{atm}/D_{atm}$ ratio is independent of the amount of
precipitation; thus, the $M_{atm}/D_{atm}$ ratio can be used as a robust index for evaluating
nitrogen saturation in forested catchments.
**1 Introduction**
Nitrate is important as a nitrogenous nutrient in the biosphere. Traditionally, forested
ecosystems have been considered as nitrogen limited (Vitousek and Howarth, 1991).
However, owing to the elevated loading of nitrogen through atmospheric deposition,
some forested ecosystems become nitrogen saturated (Aber et al., 1989), from which
elevated levels of nitrate are exported (Mitchell et al., 1997; Peterjohn et al., 1996).
Such excessive leaching of nitrate from forested catchments degrades water quality and
causes eutrophication in downstream areas (Galloway et al., 2003; Paerl and Huisman,
2009). Thus, evaluating the stage of nitrogen saturation in each forested catchment
including its temporal variation, is critical for sustainable forest management,
especially for forested ecosystems under high nitrogen deposition.
Both concentration and seasonal variation of stream nitrate have been used as indexes
to evaluate the nitrogen saturation of each forested catchment in past studies (Aber,
1992; Rose et al., 2015; Stoddard, 1994). A forested stream eluted from Fernow
Experimental Forest USA, for instance, showed an elevated average nitrate
concentration of 60 µM, along with the absence of a seasonal variation in the stream
nitrate concentration, so the forest was classified into stage 3, the highest stage of
nitrogen saturation (Rose et al., 2015).
However, using both the concentration level (high or low) and seasonal variation
(clear or absent) of stream nitrate as indexes to evaluate nitrogen saturation has
limitations, including the following (1) seasonal variation of soil nitrate can be buffered
by groundwater with long residence time, so that the seasonal variation is unclear in
stream nitrate concentration in Japan, even in normal forests under the nitrogen
saturation stage of 0 (Mitchell et al., 1997); and (2) the stream nitrate concentration can
be enriched or diluted depending on the volume of rainfall, so the concentration level
can be high in low precipitation area irrespective of the stage of nitrogen saturation.
Nakagawa et al. (2018) lately proposed that the $M_{atm}/D_{atm}$ ratio, the export flux of
unprocessed atmospheric nitrate ($M_{atm}$) relative to the deposition flux of $NO_3^-{}_{atm}$ ($D_{atm}$),
can be an alternative, more robust index for evaluating nitrogen saturation in each
forested catchment, because the $M_{atm}/D_{atm}$ ratio directly reflects the demand for
atmospheric nitrate deposited onto each forested catchments as a whole, and thus reflect
the nitrogen saturation in each forested catchment. That is, we can expect high
$M_{atm}/D_{atm}$ ratios in forested catchments under nitrogen saturation and low $M_{atm}/D_{atm}$
ratios in forested catchments with nitrogen deficiency.

To estimate the $M_{atm}/D_{atm}$ ratio accurately and precisely in each forested catchment,

the fraction of unprocessed atmospheric nitrate ($NO_3^-{}_{atm}$) in the stream needs to be
estimated accurately and precisely. Triple oxygen isotopic compositions of nitrate
($\Delta^{17}O$) have recently been used as a conservative tracer of $NO_3^-{}_{atm}$ deposited onto each
forested catchment (Inoue et al., 2021; Michalski et al., 2004; Nakagawa et al., 2018;
Tsunogai et al., 2014; Ding et al., 2022), showing distinctively different $\Delta^{17}O$ from that
of remineralized nitrate ($NO_3^-{}_{re}$), derived from organic nitrogen through general
chemical reactions, including microbial N mineralization and microbial nitrification.
While $NO_3^-{}_{re}$, the oxygen atoms of which are derived from either terrestrial $O_2$ or $H_2O$
through microbial processing (i.e., nitrification), always shows the relation close to the
"mass-dependent" relative relation between $^{17}O/^{16}O$ ratios and $^{18}O/^{16}O$ ratios; $NO_3^-{}_{atm}$
displays an anomalous enrichment in $^{17}O$ reflecting oxygen atom transfers from
atmospheric ozone ($O_3$) during the conversion of $NO_X$ to $NO_3^-{}_{atm}$ (Alexander et al.,
2009; Michalski et al., 2003; Morin et al., 2011; Nelson et al., 2018). As a result, the
$\Delta^{17}O$ signature defined by the following equation (Kaiser et al., 2007) enables us to
distinguish $NO_3^-{}_{atm}$ ($\Delta^{17}O > 0$) from $NO_3^-{}_{re}$ ($\Delta^{17}O = 0$):
$$\Delta^{17}O = \frac{1+ \delta^{17}O}{(1+ \delta^{18}O)^{\beta}} - 1 \tag{1}$$
where the constant $\beta$ is 0.5279 (Kaiser et al., 2007), $\delta^{18}O = R_{sample}/R_{standard} - 1$ and $R$ is
the $^{18}O/^{16}O$ ratio (or the $^{17}O/^{16}O$ ratio in the case of $\delta^{17}O$ or the $^{15}N/^{14}N$ ratio in the case
of $\delta^{15}N$) of the sample and each standard reference material. In addition, $\Delta^{17}O$ is almost
stable during "mass-dependent" isotope fractionation processes within terrestrial
ecosystems. Therefore, while the $\delta^{15}N$ or $\delta^{18}O$ signature of $NO_3^-{}_{atm}$ can be overprinted
by the biological processes subsequent to deposition, $\Delta^{17}O$ can be used as a robust tracer
of unprocessed $NO_3^-{}_{atm}$ to reflect its accurate mole fraction within total $NO_3^-$,
regardless of the progress of the partial metabolism (partial removal of nitrate through
denitrification and assimilation) subsequent to deposition (Michalski et al., 2004;
Nakagawa et al., 2013, 2018; Tsunogai et al., 2011, 2014, 2018).
Past studies reported that the maximum concentration of stream nitrate was 58.4 μM
in the KJ forested catchment in Japan, with the maximum value of the $M_{atm}/D_{atm}$ ratio
was 9.4 % (Nakagawa et al., 2018; Sase et al., 2022). Whether the index of the $M_{atm}/D_{atm}$
ratio can be applied to forested catchments, where the leaching of stream nitrate is much
higher than the KJ forested catchment, remained unclarified. Besides, the advantages
of the $M_{atm}/D_{atm}$ ratio within the past indexes of nitrogen saturation have not been
discussed.
Chiwa (2021) has recently reported the enrichment of nitrate of more than 90 μM on
the annual average in forested streams eluted from the FK catchments (FK1 and FK2)
in Kasuya Research Forest, Kyushu University, Japan (Figs. 1a and 1b). The observed
enrichment of stream nitrate implied that these forested catchments were under nitrogen
saturation. Thus, in this study, we determined the $M_{atm}/D_{atm}$ ratio in the FK1 and FK2
forested catchments by monitoring both the concentration and $\Delta^{17}O$ of stream nitrate
for more than 2 years to verify that these forested catchments were under nitrogen
saturation. For comparison, the MY forested catchment in Shiiba Research Forest,
Kyushu University, Japan (Figs. 1a and 1c), was also monitored during the same period,
where the average stream nitrate concentration was low (less than 10 µM). Furthermore,
the $M_{atm}/D_{atm}$ ratios in these forested catchments were compared with those reported in
past studies to verify the reliability of the $M_{atm}/D_{atm}$ ratio as an index of nitrogen
saturation.

**2 Methods**
2.1 Study sites
The FK forested catchments (33°38′N, 130°31′E) are located in a suburban area,
about 15 km west of the Fukuoka metropolitan area (the fourth largest metropolitan
area in Japan). The main plantation in these catchments was Japanese cedar/cypress
(Table 1). The MY forested catchment (32°22′N, 131°09′E) is located in a rural area at
the village of Shiiba in southern Japan's Central Kyushu Mountain range. This
catchment is a mixed forest consisting of coniferous trees such as *Abies firma Sieb. et*
*Zucc.,* and *Tsuga sieboldii Carr.,* and deciduous broadleaved trees such as *Quercus*
*crispula Blume*, *Fagus crenata Blume*, and *Acer sieboldianum* Miq. Details on the
studied forested catchments have been described in the past studies (Chiwa, 2020, 2021).

2.2 Sampling
The stream water eluted from the FK1 (14 ha), FK2 (62 ha), and MY (43 ha)
forested catchments were collected about once every month in principle from 2019/11
to 2021/12 (Fig. 1). At the FK catchments, stream water was collected at upstream
(station A) and downstream (station B) locations (Fig. 1b). At the MY catchment,
stream water was collected at station C (Fig. 1c). Samples of stream water to determine
the concentration and stable isotopic compositions ($\delta^{15}N$, $\delta^{18}O$, and $\Delta^{17}O$) of stream
nitrate were collected manually in bottles washed with deionized water before sampling
and then rinsed at least twice with the sample before sampling at each sampling site.

2.3 Analysis
All the stream water samples were passed through a membrane filter (pore size 0.45
μm) within two days after sampling and stored in a refrigerator (4 °C) until analysis.
The concentrations of nitrate were measured by ion chromatography (Prominence HIC-
SP, Shimadzu, Japan). To determine the stable isotopic compositions of nitrate in the
stream water samples, nitrate in each sample was chemically converted to $N_2O$ using a
method originally developed to determine the $^{15}N/^{14}N$ and $^{18}O/^{16}O$ ratios of seawater
and freshwater nitrate (McIlvin and Altabet, 2005) that was later modified (Konno et
al., 2010; Tsunogai et al., 2011; Yamazaki et al., 2011). In brief, 11 mL of each sample
solution was pipetted into a vial with a septum cap. Then, 0.5 g of spongy cadmium
was added, followed by 150 μL of a 1 M $NaHCO_3$ solution. The sample was then shaken
for 18-24 h at a rate of 2 cycles $s^{-1}$. Then, the sample solution (10 mL) was decanted
into a different vial with a septum cap. After purging the solution using high-purity

helium, 0.4 mL of an azide–acetic acid buffer, which had also been purged using high-purity helium, was added. After 45 min, the solution was alkalinized by adding 0.2 mL of 6 M NaOH. Then, the stable isotopic compositions ($\delta^{15}N$, $\delta^{18}O$, and $\Delta^{17}O$) of the $N_2O$ in each vial were determined using the continuous-flow isotope ratio mass spectrometry (CF-IRMS) system at Nagoya University. The analytical procedures performed using the CF-IRMS system were the same as those detailed in previous studies (Hirota et al., 2010; Komatsu et al., 2008a). The obtained values of $\delta^{15}N$, $\delta^{18}O$, and $\Delta^{17}O$ for the $N_2O$ derived from the nitrate in each sample were compared with those derived from our local laboratory nitrate standards to calibrate the values of the sample nitrate to an international scale and to correct for both isotope fractionation during the chemical conversion to $N_2O$ and the progress of oxygen isotope exchange between the nitrate derived reaction intermediate and water (ca. 20 %). In this study, we adopted the internal standard method to calibrate the stable isotopic compositions of sample nitrate. Specifically, three kinds of the local laboratory nitrate standards were used in this study, which were named to be GG01 ($\delta^{15}N$ = –3.07 ‰, $\delta^{18}O$ = +1.10 ‰, and $\Delta^{17}O$ = 0 ‰), HDLW02 ($\delta^{15}N$ = +8.94 ‰, $\delta^{18}O$ = +24.07 ‰), and NF ($\Delta^{17}O$ = +19.16 ‰), which the GG01 and the HDLW02 were used to determine the $\delta^{15}N$ and $\delta^{18}O$ of stream nitrate, and the GG01 and the NF was used to determine the $\Delta^{17}O$ of stream nitrate. The GG01, HDLW02, and NF had been calibrated using the internationally distributed isotope reference materials (USGS 34 and USGS 35). The oxygen exchange rate between nitrate and water during the chemical conversion was calculated through Eq. (2):

Oxygen exchange rate (%) = $\Delta^{17}O(N_2O)_{NF}$ / $\Delta^{17}O(NO_3^-)_{NF}$                    (2)
where the $\Delta^{17}O(N_2O)_{NF}$ denote the $\Delta^{17}O$ value of $N_2O$ that convert from the NF
nitrate, the $\Delta^{17}O(NO_3^-)_{NF}$ denote the $\Delta^{17}O$ value of NF nitrate ($\Delta^{17}O$ = +19.16 ‰)
(Tsunogai et al., 2016; Nakagawa et al., 2013, 2018; Ding et al., 2022).
The $\delta^2H$ and $\delta^{18}O$ values of $H_2O$ of the stream water samples were analyzed using
the cavity ring-down spectroscopy method by employing an L2120-i instrument
(Picarro Inc., Santa Clara, CA, USA) equipped with an A0211 vaporizer and
autosampler. The errors (standard errors of the mean) in this method were ±0.5‰ for
$\delta^2H$ and ±0.1‰ for $\delta^{18}O$. Both the VSMOW and standard light Antarctic precipitation
(SLAP) were used to calibrate the values to the international scale. The $\delta^{18}O$ values of
$H_2O$ were used to calibrate the differences in $\delta^{18}O$ of $H_2O$ between the samples and
those our local laboratory nitrate standard samples (Tsunogai et al., 2010, 2011, 2014).
To determine whether the conversion rate from nitrate to $N_2O$ was sufficient, the
concentration of nitrate in the samples was determined each time we analyzed the
isotopic composition using CF-IRMS based on the $N_2O^+$ or $O_2^+$ outputs. We adopted
the $\delta^{15}N$, $\delta^{18}O$, and $\Delta^{17}O$ values only when the concentration measured via CF-IRMS
correlated with the concentration measured via ion chromatography prior to isotope
analysis within a difference of 10 %. We repeated the analysis of $\delta^{15}N$, $\delta^{18}O$, and $\Delta^{17}O$
values for each sample at least three times to attain high precision. All samples had a
nitrate concentration of greater than 3.5 μM, which corresponded to a nitrate quantity
greater than 35 nmol in a 10 mL sample. Thus, all isotope values presented in this study
have an error (standard error of the mean) better than ±0.2 ‰ for $\delta^{15}N$, ±0.3 ‰ for $\delta^{18}O$,
and ±0.1 ‰ for $\Delta^{17}O$.
Nitrite ($NO_2^-$) in the samples interferes with the final $N_2O$ produced from nitrate
because the chemical method also converts $NO_2^-$ to $N_2O$ (McIlvin and Altabet, 2005).
Therefore, it is sometimes necessary to remove $NO_2^-$ prior to converting nitrate to $N_2O$.
In this study, however, we skipped the processes for removing $NO_2^-$ because all the
stream samples analyzed for stable isotopic composition had $NO_2^-$ concentrations lower
than the detection limit (0.05 µM).

2.4 Deposition rate of atmospheric nitrate
The annual deposition rate of atmospheric nitrate ($D_{atm}$; total dry and wet deposition
rate of atmospheric nitrate) in each catchment was estimated using the annual "bulk"
deposition rate of atmospheric nitrate ($D_{bulk}$) calculated in Chiwa (2020) at each
catchment by multiplying the volume-weighted mean concentration of nitrate in the
bulk deposition samples collected every 2 weeks at each catchment for 10 years (from
2009/1 to 2018/12) by the annual amount of precipitation. The bulk deposition samples
were those accumulated in a plastic bucket installed in an open site of each catchment
55 cm above the ground. The distances between the monitoring sites of bulk deposition
in the FK1, FK2, and MY forested catchments and the stations of stream water sampling
(stations A, B, and C) were 3.9, 2.9, and 4.5 km, respectively. The concentrations of
nitrate in the bulk deposition samples were measured by ion chromatography.
The $D_{bulk}$ determined through this method, however, is less than $D_{atm}$ (Aikawa et al.,
2003) because the dry deposition velocities of gases and particles on the water surface
of the plastic bucket are smaller than those on the forest (Matsuda, 2008). Thus, we
corrected the differences by using Eq. (3) to estimate $D_{atm}$ from $D_{bulk}$:
$D_{atm} = D_{bulk} - D_{dry}(W) + D_{dry}(F)$                                   (3)
where $D_{dry}(W)$ and $D_{dry}(F)$ denote the annual dry deposition rates onto water and forest,
respectively.
The $D_{dry}(W)$ and $D_{dry}(F)$ at each catchment were determined using an inferential
method (Endo et al., 2011) through Eqs. (4) and (5), respectively:
$D_{dry}(W) = [NO_{3\ atm}^-]_{gas} \times V_{gas}(W) + [NO_{3\ atm}^-]_p \times V_p(W)$     (4)
$D_{dry}(F) = [NO_{3\ atm}^-]_{gas} \times V_{gas}(F) + [NO_{3\ atm}^-]_p \times V_p(F)$      (5)
where $[NO_{3\ atm}^-]_{gas}$ denotes the concentration of gaseous nitrate in air; $[NO_{3\ atm}^-]_p$
denotes the concentration of particle nitrate in air; $V_{gas}(W)$ and $V_{gas}(F)$ denote the
deposition velocities of gaseous nitrate on the water surface and forest, respectively;
and $V_p(W)$ and $V_p(F)$ denote the deposition velocities of particulate nitrate on the water
surface and forest, respectively. Those determined by Chiwa (2010) using the annular
denuder method from 2006/5 to 2007/4 were used for the $[NO_3^-]_{gas}$ and $[NO_3^-]_p$ in the
FK catchments. Those determined by the National Institute for Environmental Studies
(Environmental Laboratories Association of Japan, 2017) using the filter-pack method
at Miyazaki (31°83′N, 131°42′E) from 2011 to 2017 were used for the $[NO_3^-]_{gas}$ and
$[NO_3^-]_p$ in the MY catchment. The $V_{gas}(F)$, $V_{gas}(W)$, $V_p(F)$, and $V_p(W)$ of each
catchment were determined by applying the estimation file for dry deposition (Matsuda,

2008;

http://www.hro.or.jp/list/environmental/research/ies/katsudo/acid_rain/kanseichinchak
u/kanseichinchaku.html), where $V_{gas}$ and $V_p$ were calculated using the meteorological
data of wind speed, temperature, humidity, radiation, and cloud amount and land use.
The meteorological data monitored by Japan Meteorological Agency at the nearest
Fukuoka station (33°34′N, 130°22′E) and Miyazaki station (31°56′N, 131°24′E) from
2009 to 2021 were used for the FK and MY catchments, respectively. The forested land
use of 100 % was chosen for each area.

2.5 Flux of stream water
The flux of stream water ($F_{stream}$) in each catchment was not measured fully in this
study. Instead, the water balance in each catchment was used to estimate $F_{stream}$,
assuming that the outflux of water from the study catchments to deep groundwater was
negligible:
$F_{stream} = P - E$ (6)
where P denotes the annual average precipitation and E denotes the annual
evapotranspiration flux of water in each catchment. In this paper, the equation obtained
by Komatsu et al. (2008) was used to estimate the E of the FK and MY catchments.
Details on this equation are shown below.
Komatsu et al. (2008) compiled the annual flux of evapotranspiration determined in
43 forested catchments in Japan and found that E shows a positive correlation with the
average temperature ($T_{avg}$) of each catchment. Thus, they proposed the modeled relation
of E (mm) = 31.4$T_{avg}$ (°C) + 376 to estimate E in each forested catchment in Japan,
where the standard error of 162.3 mm was included in the estimated evapotranspiration
flux (E). They also confirmed that the estimated $F_{stream}$ using the model corresponded
well with the observed $F_{stream}$ in three forested catchments, with estimated errors of less
than 6 %. As a result, we utilized the water balance method proposed by Komatsu et al.
(2008) to quantify the $F_{stream}$ in each catchment.

2.6 Concentration of unprocessed $NO_3^-{}_{atm}$ in each water sample
The $\Delta^{17}O$ data of nitrate in each sample was used to estimate the concentration of
$NO_3^-{}_{atm}$ ([$NO_3^-{}_{atm}$]) in each water sample by applying Eq. (7):
$[NO_3^-{}_{atm}]/[NO_3^-] = \Delta^{17}O/\Delta^{17}O_{atm}$                                        (7)
where [$NO_3^-{}_{atm}$] and [$NO_3^-$] denote the concentrations of $NO_3^-{}_{atm}$ and nitrate (total) in
each water sample, respectively, and $\Delta^{17}O_{atm}$ and $\Delta^{17}O$ denote the $\Delta^{17}O$ values of
$NO_3^-{}_{atm}$ and nitrate (total) in the stream water sample, respectively. In this study, we
used the annual average $\Delta^{17}O$ value of $NO_3^-{}_{atm}$ determined at the Sado-Seki monitoring
station in Japan (Sado Island; Fig. 1a) from April 2009 to March 2012 ($\Delta^{17}O_{atm}$ =
+26.3 ‰; Tsunogai et al., 2016) for $\Delta^{17}O_{atm}$ in Eq. (7) to estimate [$NO_3^-{}_{atm}$] in the stream.
We allow for an error range of 3 ‰ in $\Delta^{17}O_{atm}$, where the factor changes in $\Delta^{17}O_{atm}$
from +26.3 ‰ caused by both areal and seasonal variations in the $\Delta^{17}O$ values of
$NO_3^-{}_{atm}$ have been considered (Nakagawa et al., 2018; Tsunogai et al., 2016; Ding et
al., 2022).

The annual export flux of unprocessed $NO_3^-{}_{atm}$ per unit area of the catchment ($M_{atm}$)

was determined by applying Eq. (8):
$M_{atm} = [NO_3^-{}_{atm}]_{avg} \times F_{stream}$                                           (8)
where $[NO_3^-{}_{atm}]_{avg}$ denotes the annual average $[NO_3^-{}_{atm}]$ in each stream. The index of
nitrogen saturation ($M_{atm}/D_{atm}$ ratio) was calculated by dividing $M_{atm}$ with $D_{atm}$ in each
catchment.

2.7 Concentration and isotopic compositions of stream nitrate eluted only from the FK2
catchment

The concentration and isotopic compositions ($\delta^{15}N$, $\delta^{18}O$, and $\Delta^{17}O$) of stream nitrate

determined at station B were the mixtures of those eluted from FK1 and FK2
catchments (Fig. 1b). Assuming that the stream nitrate eluted from FK1 catchment was
stable during the flow path from station A to station B. The concentration of stream
nitrate eluted from the FK2 catchment was determined by applying Eq. (9):
$[NO_3^-]_{FK2} = ([NO_3^-]_{FK1+FK2} * F_{FK1+FK2} - [NO_3^-]_{FK1} * F_{FK1}) / F_{FK2}$         (9)
where $F_{FK1}$, $F_{FK2}$, and $F_{FK1+FK2}$ denote the flux of stream water eluted from the FK1,
FK2 (only), and FK1+FK2 catchment, respectively. $[NO_3^-]_{FK1}$, $[NO_3^-]_{FK2}$, and
$[NO_3^-]_{FK1+FK2}$ denote the concentration of stream nitrate eluted from the FK1, FK2
(only), and FK1+FK2 catchment, respectively. In this study, the flow rates measured at
stations A and B on 2021/01/15 by using the salt dilution method (Sappa et al., 2015)
was used for $F_{FK1}$ (0.85 L/s) and $F_{FK1+FK2}$ (4.75 L/s), respectively, and the measured
$[NO_3^-]$ at stations A and B was used for $[NO_3^-]_{FK1}$ and $[NO_3^-]_{FK1+FK2}$, respectively.
Because the relation between the measured flow rates was comparable with the relation
between the catchment area of FK1 (14 ha) and that of FK1+FK2 (76 ha), we concluded
that the measured flow rates of 0.85 L/s and 4.75 L/s were reasonable as for those
representing the $F_{FK1}$ and $F_{FK1+FK2}$, respectively. According to the mass balance of water,
we can estimate the $F_{FK2}$ eluted from the FK2 catchment only to be 3.90 L/s.
Assuming that the stream nitrate eluted from FK1 catchment was stable during the
flow path from station A to station B, the $\delta^{15}N$, $\delta^{18}O$, and $\Delta^{17}O$ values of stream nitrate
eluted from the FK2 catchment only were determined by applying Eq. (10):
$\delta_{FK2} = (\delta_{FK1+FK2} * [NO_3^-]_{FK1+FK2} * F_{FK1+FK2} - \delta_{FK1} * [NO_3^-]_{FK1} * F_{FK1}) / ([NO_3^-]_{FK2} *$
$F_{FK2})$                                                                                                            (10)
where $\delta_{FK1}$, $\delta_{FK2}$, and $\delta_{FK1+FK2}$ denote the $\delta^{15}N$ (or $\delta^{18}O$ or $\Delta^{17}O$) of stream nitrate eluted
from the FK1, FK2, and FK1+FK2 catchment, respectively. The $\delta^{15}N$ (or $\delta^{18}O$ or $\Delta^{17}O$)
values of stream nitrate measured at stations A and B were used for $\delta_{FK1}$ and $\delta_{FK1+FK2}$,
respectively.

**3 Results**
3.1 Deposition rate of atmospheric nitrate
The mean annual precipitation (P) from 2009 to 2021 was 1777 mm and 3981 mm

for FK and MY catchments, respectively (Chiwa, 2020; Chiwa, personal communication, September 21, 2022). The mean annual temperature ($T_{avg}$) was reported to be 15.9 °C and 10.8 °C for FK and MY catchments, respectively (Chiwa, 2020). Based on these data, the annual flux of stream water ($F_{stream}$) was estimated to be 902.0 ± 162.3 mm at FK catchments and 3266.1 ± 162.3 mm at MY catchment, respectively, using Eq. (6).

Chiwa (2020) reported the annual bulk deposition rates of atmospheric nitrate ($D_{bulk}$) to be 34.0 mmol m$^{-2}$ year$^{-1}$ at FK catchments and 24.2 mmol m$^{-2}$ year$^{-1}$ at MY catchment. On the other hand, the annual dry deposition rate of atmospheric nitrate ($D_{dry}$) deposited on the forest ($D_{dry}$(F)) and on the water surface ($D_{dry}$(W)) were estimated to be 39.9 mmol m$^{-2}$ year$^{-1}$ and 4.1 mmol m$^{-2}$ year$^{-1}$, respectively, at FK catchments, and 18.4 mmol m$^{-2}$ year$^{-1}$ and 2.4 mmol m$^{-2}$ year$^{-1}$, respectively, at MY catchment. As a result, $D_{atm}$ was estimated to be 69.3 mmol m$^{-2}$ year$^{-1}$ at FK catchments and 40.1 mmol m$^{-2}$ year$^{-1}$ at MY catchment, using Eq. (3).

3.2 Concentration and isotopic composition of stream nitrate

The concentrations of stream nitrate eluted from the FK1, FK2 (only), and MY catchments ranged from 97.5 µM to 121.3 µM, from 65.7 µM to 148.5 µM, and from 3.5 µM to 15.3 µM, respectively, with the average concentrations of 109.5 µM, 90.9 µM, and 7.3 µM, respectively, and the standard deviations (SD) of 6.3 µM, 18.5 µM, and 3.0 µM, respectively, which corresponds to the coefficients of variation (CV) of

5.7 %, 20.4 %, and 40.7 %, respectively (Fig. 2a). All catchments showed no clear
seasonal variation during the observation periods. The variation ranges and the average
concentrations of stream nitrate eluted from the three catchments agreed well with the
past observations performed in the same catchments (Chiwa, 2021).
The stable isotopic compositions of stream nitrate eluted from the FK1, FK2 (only),
and MY catchments ranged from −0.9 ‰ to +1.5 ‰, from −1.4 ‰ to +5.8 ‰, and from
−0.8 ‰ to +2.4 ‰, respectively, for $\delta^{15}N$ (Fig. 2b), from +3.9 ‰ to +8.5 ‰, from −2.2 ‰
to +2.8 ‰, and from −5.6 ‰ to +1.7 ‰, respectively, for $\delta^{18}O$ (Fig. 2c), and from +2.0 ‰
to +3.3 ‰, from +0.6 ‰ to +2.2 ‰, and from +0.2 ‰ to +1.0 ‰, respectively, for $\Delta^{17}O$
(Fig. 2d), with no clear seasonal variation during the observation periods. The
concentration-weighted averages for the $\delta^{15}N$, $\delta^{18}O$, and $\Delta^{17}O$ values of stream nitrate
were +0.2 ‰, +6.4 ‰, and +2.6 ‰, respectively, at FK1, +1.0 ‰, +0.5 ‰, and +1.5 ‰,
respectively, at FK2, +0.7 ‰, −2.5 ‰, and +0.6 ‰, respectively, at MY.

3.3 Concentration of unprocessed atmospheric nitrate and the $M_{atm}/D_{atm}$ ratio in each
catchment
The concentration of unprocessed atmospheric nitrate ($[NO_3^-{}_{atm}]$) in the streams eluted
from the FK1, FK2 (only), and MY catchments ranged from 8.64 to 14.30 μM, from
2.27 to 10.71 μM, and from 0.03 to 0.46 μM with the average concentration of 10.80 ±
1.30, 5.06 ± 0.67, and 0.16 ± 0.03 μM, respectively, even though these studied
catchments showed little seasonal variations during the observation periods (Fig. 2e).
The annual export flux of nitrate ($M_{total}$), the annual export flux of $NO_3^-{}_{atm}$ ($M_{atm}$), and
the $M_{atm}/D_{atm}$ ratio were $98.8 \pm 17.8$ mmol m$^{-2}$ year$^{-1}$, $9.7 \pm 2.1$ mmol m$^{-2}$ year$^{-1}$, and
$14.1 \pm 4.1$ % at FK1 catchment, respectively, $82.0 \pm 14.8$ mmol m$^{-2}$ year$^{-1}$, $4.6 \pm 1.0$
mmol m$^{-2}$ year$^{-1}$, and $6.6 \pm 2.0$ % at FK2 catchment, respectively, $23.7 \pm 1.2$ mmol m$^{-}$
$^2$ year$^{-1}$, $0.5 \pm 0.1$ mmol m$^{-2}$ year$^{-1}$, and $1.3 \pm 0.4$ % at MY catchment, respectively
(Table 2). The uncertainties of $[NO_3^-{}_{atm}]$, $M_{atm}$, and $M_{atm}/D_{atm}$ ratio in each catchment
were determined from the uncertainties of $\Delta^{17}O$, $\Delta^{17}O_{atm}$, $F_{stream}$, and $D_{atm}$ according to
the equations of error propagation. The details were described in Appendix A.

**4 Discussion**
4.1 Deposition rate of atmospheric nitrate at the study catchments

Based on the air monitoring data determined at the stations of Fukuoka (33°51′N,

130°50′E) and Miyazaki (31°83′N, 131°42′E) from 2011 to 2017, the Environmental
Laboratories Association of Japan (2017) reported $D_{atm}$ to be 57.8 mmol m$^{-2}$ year$^{-1}$ at
Fukuoka and 49.1 mmol m$^{-2}$ year$^{-1}$ at Miyazaki. Those values are consistent with the
$D_{atm}$ estimated in this study (69.3 and 40.1 mmol m$^{-2}$ year$^{-1}$ at the FK and MY
catchments, respectively), within a difference of approximately 20 %. Thus, we
concluded that the $D_{atm}$ estimated in this study was reliable within the error margin of
20 % (Table 2). Because the $D_{atm}$ determined at the FK catchments was the highest
among the forested catchments in Table 3, we further compared the $D_{atm}$ of the FK
catchments with those from the other air monitoring stations in Japan reported in past
studies, along with that of the MY catchment (Table S1). While the $D_{atm}$ of the MY
catchment corresponded to the average level among the sites compiled in Table S1, the
$D_{atm}$ of the FK catchments exceeded the average level significantly. In addition, the $D_{atm}$
of the FK catchments corresponded to one of the highest among the Japanese forested
areas (Table S1). All the catchments in Japan can be suffered from the long-range
transport of air pollutants derived from megacities in East Asian region (Chiwa, 2021;
Chiwa et al., 2012 and 2013).In addition, the shorter transport distance from the
Fukuoka metropolitan area (total population: 1.62 million people; population density:
4715 people/km$^2$) may be mainly responsible for the $D_{atm}$ higher in FK than in MY,
because the FK catchments are only 15 km west of the Fukuoka metropolitan area.

4.2 Excess leaching of unprocessed atmospheric nitrate from FK catchments

The isotopic compositions ($\delta^{15}N$, $\delta^{18}O$, and $\Delta^{17}O$) of stream nitrate eluted from the

FK and MY catchments were typical for those eluted from forested catchments (Hattori
et al., 2019; Huang et al., 2020; Nakagawa et al., 2013, 2018; Riha et al., 2014; Sabo et
al., 2016; Tsunogai et al., 2014, 2016). The striking features found in the FK catchments
were that, in addition to the high [$NO_3^-$] and high $M_{total}$ that had been clarified in a past
study (Chiwa, 2021), both [$NO_{3\ atm}^-$] and $M_{atm}$ in FK were higher than those eluted from
MY (Table 2). Especially, the average [$NO_{3\ atm}^-$] in the stream eluted from the FK1
catchment was the highest ever reported in forested streams determined through
continuous monitoring for more than 1 year (Bostic et al., 2021; Bourgeois et al., 2018b,
2018a; Hattori et al., 2019; Huang et al., 2020; Nakagawa et al., 2018; Rose et al., 2015;
Sabo et al., 2016; Tsunogai et al., 2014, 2016).

The observed high $[NO_3^-{}_{atm}]$ in the stream eluted from the FK1 catchment could be

caused just by the high $[NO_3^-{}_{atm}]$ deposition in the catchment. Thus, we compiled all
past data ever reported in forested streams through continuous monitoring in Table 3,
where the data of average $[NO_3^-]$, average $[NO_3^-{}_{atm}]$, $M_{atm}$, $M_{total}$, $D_{atm}$, and $M_{atm}/D_{atm}$
ratio were included for comparison. The result showed that the $M_{atm}/D_{atm}$ ratio, along
with $M_{atm}$, was the highest as well in the FK1 catchment among the forested catchments
(Table 3).

Elevated loading of nitrogen through atmospheric deposition was responsible for the

occurrence of nitrogen saturation in forest ecosystems, from which elevated levels of
nitrate are exported (Aber et al., 1989). Nakagawa et al. (2018) proposed that the
$M_{atm}/D_{atm}$ ratio can be an index for evaluating the nitrogen saturation in each forested
catchment, because the $M_{atm}/D_{atm}$ ratio directly reflects the present demand for
atmospheric nitrate deposited in each forested catchment, and thus reflects the nitrogen
saturation in each forested catchment. The high $M_{atm}/D_{atm}$ ratios observed in the FK
catchments implied that the demand for atmospheric nitrate was low in the FK
catchments and that the stages of nitrogen saturation at the FK catchments were higher
than those at other forested catchments. That is, the nitrogen saturation at the FK
catchments was responsible for the observed high $[NO_3^-]$ and high $M_{total}$ at the FK
catchments than at MY and any other catchment ever studied (Table 3).
The stand age of forests can affect the retention or loss of N (Fukushima et al., 2011;
Ohrui and Mitchell, 1997). Fukushima et al. (2011) evaluated N uptake rates of
Japanese cedars at different ages (5-89 years old) and demonstrated that the N uptake
rates of Japanese cedars were higher in younger stands (53 kg N ha$^{-1}$ year$^{-1}$ in 16 years
old) than in older stands (29 kg N ha$^{-1}$ year$^{-1}$ in 31 years old; 24 kg N ha$^{-1}$ year$^{-1}$ in 42
years old; 34 kg N ha$^{-1}$ year$^{-1}$ in 89 years old). In addition, Yang and Chiwa (2021)
found that the nitrate concentration in the soil water taken beneath the rooting zone of
matured artificial Japanese cedar plantations (607 $\pm$ 59 μM; 64-69 years old) was
significantly higher than that of normal Japanese oak plantations (8.7 $\pm$ 8.1 μM; 24
years old). Moreover, by adding ammonium nitrate (50 kg N ha$^{-1}$ year$^{-1}$) to the forest
floor directly, Yang and Chiwa (2021) found that the nitrate concentration in the soil
water of the matured artificial Japanese cedar plantations increased significantly faster
than that of the normal Japanese oak plantations, probably because of the lower N
uptake rates in the matured artificial Japanese cedar plantations. Because most of the
artificial Japanese cedar/cypress plantations in the FK and MY catchments have reached
their maturity (> 50 years; Yang and Chiwa, 2021), the higher proportion of matured
artificial Japanese cedar/cypress plantations in the FK1 catchment (Table 1) was highly
responsible for the observed elevated leaching of nitrate, caused by the reduction in N
uptake rates.
As a result, we concluded that the FK forested catchments were under the high
nitrogen saturation stage, FK1 catchment especially, and the nitrogen saturation in the
FK1 catchment was responsible for the elevated $M_{total}$, $M_{atm}$, $[NO_3^-]$, $[NO_{3\ atm}^-]$ found
in the stream eluted from the catchment (Figs. 3a, 3b, 3c and 3d).

4.3 The $M_{atm}/D_{atm}$ ratio as an index of nitrogen saturation
Past studies have used the concentration of stream nitrate as one of the important
indexes to evaluate the stage of nitrogen saturation in each forest (Aber, 1992; Huang
et al., 2020; Rose et al., 2015; Stoddard, 1994). The strong linear relationship ($R^2 = 0.76$;
$P < 0.0001$) between the stream nitrate concentration and the $M_{atm}/D_{atm}$ ratio, except
for the Qingyuan forested catchment (Fig. 3d), further supported that the $M_{atm}/D_{atm}$ ratio
can be used as an alternative index of nitrogen saturation, as pointed out in Nakagawa
et al. (2018).
The differences in the number of storm and/or snowmelt events could affect the
$M_{atm}/D_{atm}$ ratio as well, because $NO_{3\ atm}^-$ could be injected into the stream water directly,
along with the storm/snowmelt water (Tsunogai et al., 2014; Ding et al., 2022; Inamdar
and Mitchell, 2006). In recent study, however, we found that storm events have little
impact on the $M_{atm}/D_{atm}$ ratio, based on monitoring temporal variation of $[NO_{3\ atm}^-]$ in
stream water during storm events (Ding et al., 2022). In addition, the low $M_{atm}/D_{atm}$
ratio found in Uryu forested catchment (0.7 %; Table 3) implied that the snowmelt has
little impact on the $M_{atm}/D_{atm}$ ratio as well, because 30% of the annual mean
precipitation was snow in Uryu forested catchment (Tsunogai et al., 2014).
The differences in the amount of precipitation, temperature, and the flux of stream
water could affect the $M_{atm}/D_{atm}$ ratio as well. As a result, the annual amount of
precipitation, mean temperature, and the annual mean flux of stream water ($F_{stream}$) in
the forested catchments were compiled in Table S2. While the stream nitrate
concentration showed a strong linear relationship ($R^2 = 0.76$; $P < 0.0001$) with the
$M_{atm}/D_{atm}$ ratio (Fig. 3d), the precipitation, temperature, and $F_{stream}$ did not show a
significant relationship with the $M_{atm}/D_{atm}$ ratio ($P > 0.14$; Fig. 4). As a result, we
concluded that the $M_{atm}/D_{atm}$ ratio was mainly controlled by the progress of nitrogen
saturation, rather than the differences in the number of storm and/or snowmelt events,
the amount of precipitation, temperature, and the flux of stream water.
The differences in the residence time of water in each catchment could also impact
the $M_{atm}/D_{atm}$ ratio, as the residence time of water in forested catchments ranges from
one month to more than one year (Asano et al., 2002; Farrick and Branfireun, 2015;
Kabeya et al., 2008; Rodgers et al., 2005; Soulsby et al., 2006; Tetzlaff et al., 2007). It
is difficult to explain high [$NO_3^-$] and high $M_{total}$ eluted from the catchment by the
residence time of water alone, while the $M_{atm}/D_{atm}$ ratio could be higher in catchments
with a shorter water residence time, as the majority of nitrate eluted from the catchment
with a high $M_{atm}/D_{atm}$ ratio was $NO_{3\ re}^-$ produced by microbial nitrification. The
significant correlation between $M_{total}$ and $M_{atm}/D_{atm}$ ratios ($P < 0.0001$; Fig. 3a)
supported nitrogen saturation as the leading cause of high $M_{total}$ in catchments with a
high $M_{atm}/D_{atm}$ ratio. Additionally, the high loading of atmospheric nitrogen, the type of
plantation, and the old age of plantation in the FK1 catchment all supported the
conclusion that the FK1 catchment was under nitrogen saturation.
The $M_{atm}/D_{atm}$ ratio is a more reliable and robust index than the stream nitrate
concentration, as explained below. The Qingyuan forested catchment can be classified
into the highest nitrogen saturation stage based only on the highest stream nitrate
concentration of 150 μM (Table 3). However, based on the leaching flux of nitrogen via
stream water monitored by Huang et al. (2020) for 4 years in the Qingyuan forested
catchment, along with the deposition flux of nitrogen, we can obtain the $M_{atm}/D_{atm}$ ratio
in the catchment to be a medium level of 5.8 ± 1.3 %, implying that the nitrogen
saturation stage was not so high (Table 3). Huang et al. (2020) also concluded that the
input of nitrogen exceeded the output in the catchment, and thus, the catchment was at
stage 2 of nitrogen saturation. The $M_{atm}/D_{atm}$ ratio in the Qingyuan forested catchment
with a medium level among all forested catchments (Fig. 3d) should be a more reliable
index of nitrogen saturation.

Compared with those in the other forested catchments in Table 3, the annual amount

of precipitation (P) has the lowest value of 709 mm in the Qingyuan forested catchment.
The flux of stream water ($F_{stream}$) has the lowest value of 309 mm as well. Thus, we
concluded that nitrate was relatively concentrated in the catchment because of the small
precipitation, resulting in relative enrichment in the concentrations of both nitrate (150
μM) and unprocessed atmospheric nitrate (8.9 μM) in the stream.

While the concentration of stream nitrate, as an index of nitrogen saturation

traditionally, can be influenced by the amount of precipitation, as demonstrated in the
Qingyuan forested catchment, the $M_{atm}/D_{atm}$ ratio is independent of the amount of
precipitation (Fig. 4). Therefore, the $M_{atm}/D_{atm}$ ratio can be used as a more robust index
for evaluating nitrogen saturation in each forested catchment.

**5 Conclusions**
Both the concentrations and $\Delta^{17}O$ of stream nitrate were determined for more than 2
years in the forested catchments of FK (FK1 and FK2) and MY to determine the
$M_{atm}/D_{atm}$ ratio for each catchment. The FK catchments exhibited higher $M_{atm}/D_{atm}$ ratio
than the MY catchment and other forested catchments reported in past studies, implying
that the progress of nitrogen saturation in the FK catchments was severe. Both age and
proportion of artificial plantation in the FK catchments were responsible for the
progress of nitrogen saturation. In addition, although past studies have commonly used
the concentration of stream nitrate as an index to evaluate the progress of nitrogen
saturation in forested catchments, it can be influenced by the amount of precipitation.
As a result, we concluded that the $M_{atm}/D_{atm}$ ratio should be used as a more reliable
index for evaluating the progress of nitrogen saturation because the $M_{atm}/D_{atm}$ ratio is
independent from the amount of precipitation.

Appendix A: Calculating of uncertainties in the values of $[NO_3^-{}_{atm}]$, $M_{atm}$, and $M_{atm}/D_{atm}$
ratio
The uncertainty in the values of $[NO_3^-{}_{atm}]$ was estimated from the uncertainties in
the $\Delta^{17}O$ values of stream nitrate ($\Delta^{17}O$) and $NO_3^-{}_{atm}$ ($\Delta^{17}O_{atm}$) according to the
divisive equation of error propagation (A1):
$$\sigma_{[NO_3^-{}_{atm}]} = [NO_3^-] * \sqrt{\left(\frac{1}{\Delta^{17}O_{atm}} * \sigma_{\Delta^{17}O}\right)^2 + \left(\frac{\Delta^{17}O}{\Delta^{17}O_{atm}^2} * \sigma_{\Delta^{17}O_{atm}}\right)^2} \quad\quad (A1)$$
where $\sigma_{[NO_3^-{}_{atm}]}$, $\sigma_{\Delta^{17}O}$, and $\sigma_{\Delta^{17}O_{atm}}$ denote the uncertainties in $[NO_3^-{}_{atm}]$, $\Delta^{17}O$ values
of stream nitrate, and $\Delta^{17}O$ values of $NO_3^-{}_{atm}$, respectively. The standard error of the
mean (SE) of ±0.1 ‰ and the areal/seasonal variations of ±3 ‰ was used in
calculating $\sigma_{\Delta^{17}O}$ and $\sigma_{\Delta^{17}O_{atm}}$, respectively. As a result, the uncertainty in $[NO_3^-{}_{atm}]$
$(\sigma_{[NO_3^-{}_{atm}]})$ was ±1.30, ±0.67, and ±0.03 μM at FK1, FK2, and MY catchments,
respectively.

The uncertainty in the values of $M_{atm}$ was estimated from the uncertainties in

$[NO_3^-{}_{atm}]$ and in $F_{stream}$ according to the multiplicative equation of error propagation
(A2):
$$\sigma_{M_{atm}} = \sqrt{\left(F_{stream} * \sigma_{[NO_3^-{}_{atm}]}\right)^2 + \left([NO_3^-{}_{atm}] * \sigma_{F_{stream}}\right)^2} \quad\quad (A2)$$
where $\sigma_{M_{atm}}$, $\sigma_{[NO_3^-{}_{atm}]}$, and $\sigma_{F_{stream}}$ denote the uncertainties in $M_{atm}$, $[NO_3^-{}_{atm}]$, and
$F_{stream}$, respectively. Komatsu et al. (2008) proposed the uncertainty in $F_{stream}$ to be
±162.3 mm when using the water balance method in estimating $F_{stream}$. Here, the
uncertainty in $M_{atm}$ ($\sigma_{M_{atm}}$) was ±2.1, ±1.0, and ±0.1 mmol m$^{-2}$ yr$^{-1}$ at FK1, FK2, and
MY catchments, respectively.

The uncertainty in $M_{atm}/D_{atm}$ ratio was estimated from the uncertainties in $M_{atm}$ and

in $D_{atm}$ according to the divisive equation of error propagation (A3):
$$\sigma_{M_{atm}/D_{atm}\ ratio} = \sqrt{\left(\frac{1}{D_{atm}} * \sigma_{M_{atm}}\right)^2 + \left(\frac{M_{atm}}{D_{atm}^2} * \sigma_{D_{atm}}\right)^2} \quad\quad (A3)$$
where $\sigma_{M_{atm}/D_{atm}\ ratio}$, $\sigma_{M_{atm}}$, and $\sigma_{D_{atm}}$ denote the uncertainty in $M_{atm}/D_{atm}$ ratio,
$M_{atm}$, and $D_{atm}$, respectively. Comparing the deposition rate of $NO_3^-{}_{atm}$ obtained at the
other atmospheric monitoring stations nearby, the uncertainty of 20 % was adopted
for those of $D_{atm}$ in each catchment, which corresponds to the uncertainty in $D_{atm}$ of
±13.9, ±13.9, ±8.0 mmol m$^{-2}$ yr$^{-1}$ at FK1, FK2, and MY catchments, respectively. As
a result, the uncertainty in $M_{atm}/D_{atm}$ ratio was ±4.1 %, ±2.0 %, and ±0.4 % at FK1,
FK2, and MY catchments, respectively.

*Data availability.* All the primary data are presented in the Supplement. The other data
are available upon request to the corresponding author (Weitian Ding).

*Author contributions.* UT, FN, KS, and MC designed the study. MC and TK performed
the field observations. WD, UT, and FN determined the concentrations and isotopic
compositions of the samples. WD, TS, FN, and UT performed data analysis, and WD
and UT wrote the paper with input from MC, TK, and KS.

*Competing interests.* The authors declare that they have no conflict of interest.

*Acknowledgements.*
We thank anonymous referees for valuable remarks on an earlier version of this paper.
We also thank Daisuke Nanki, Takuma Nakamura, and Yuko Muramatsu for their
long-term water sampling. Additionally, we are grateful to the members of the
Biogeochemistry Group, Graduate School of Environmental Studies, Nagoya
University, for their valuable support throughout this study. This work was supported
by a Grant-in-Aid for Scientific Research from the Ministry of Education, Culture,
Sports, Science, and Technology of Japan under grant numbers 22H00561, and
17H00780, the Yanmar Environmental Sustainability Support Association, and the
River fund of the river foundation, Japan. Weitian Ding would like to take this
opportunity to thank the "Nagoya University Interdisciplinary Frontier Fellowship"
supported by Nagoya University and JST, the establishment of university fellowships
towards the creation of science technology innovation, Grant Number JPMJFS2120.

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

**Table 1.** Plant information for each forested catchment (Chiwa, 2021).

| Overstory vegetation (%) | FK1 | FK2 | MY |
|---|---|---|---|
| Artificial Japanese cedar/cypress plantation | 74 | 40 | 16 |
| Other artificial coniferous plantations | <1 | <1 | 7 |
| Natural trees | 10 | 54 | 75 |
| Others | 16 | 5 | 2 |




**Table 2.** Average concentrations of stream nitrate ($[NO_3^-]_{avg}$), the average
concentrations of unprocessed $NO_3^-{}_{atm}$ in streams ($[NO_3^-{}_{atm}]_{avg}$), the annual export flux
of $NO_3^-$ per unit area of catchments ($M_{total}$), the annual export flux of $NO_3^-{}_{atm}$ per unit
area of catchments ($M_{atm}$), the deposition flux of $NO_3^-{}_{atm}$ per unit area of catchment
($D_{atm}$), and the $M_{atm}/D_{atm}$ ratios in the study catchments.

| | FK1 | FK2 | MY |
|---|---|---|---|
| $[NO_3^-]_{avg}$ (μM) | 109.5 | 90.9 | 7.3 |
| $[NO_3^-{}_{atm}]_{avg}$ (μM) | $10.80 \pm 1.30$ | $5.06 \pm 0.67$ | $0.16 \pm 0.03$ |
| $M_{total}$ (mmol m$^{-2}$ yr$^{-1}$) | $98.8 \pm 17.8$ | $82.0 \pm 14.8$ | $23.7 \pm 1.2$ |
| $M_{atm}$ (mmol m$^{-2}$ yr$^{-1}$) | $9.7 \pm 2.1$ | $4.6 \pm 1.0$ | $0.5 \pm 0.1$ |
| $D_{atm}$ (mmol m$^{-2}$ yr$^{-1}$) | $69.3 \pm 13.9$ | $69.3 \pm 13.9$ | $40.1 \pm 8.0$ |
| $M_{atm}/D_{atm}$ (%) | $14.1 \pm 4.1$ | $6.6 \pm 2.0$ | $1.3 \pm 0.4$ |


**Table 3.** The annual amount of precipitation (P), the average concentration of stream
nitrate ($[NO_3^-]_{avg}$), the nitrogen saturation stage, the average concentration of
unprocessed $NO_3^-{}_{atm}$ in streams ($[NO_3^-{}_{atm}]_{avg}$), the annual export flux of $NO_3^-$ per unit
area of catchment ($M_{total}$), the annual export flux of $NO_3^-{}_{atm}$ per unit area of catchment
($M_{atm}$), the deposition flux of $NO_3^-{}_{atm}$ per unit area of catchment ($D_{atm}$), and the
$M_{atm}/D_{atm}$ ratio in the FK1, FK2, and MY, along with those in the catchments studied in
past studies using $\Delta^{17}O$ of nitrate as a tracer.

| | P | $[NO_3^-]_{avg}$ | N stage[*] | $[NO_3^-{}_{atm}]_{avg}$ | $M_{atm}$ | $M_{total}$ | $D_{atm}$ | $M_{atm}/D_{atm}$ |
|---|---|---|---|---|---|---|---|---|
| | mm | μM | | μM | mmol m$^{-2}$ yr$^{-1}$ | | | % |
| FK1[a] | 1777 | 109.5 | - | 10.8 | 9.7 | 98.8 | 69.3 | 14.1 |
| FK2[a] | 1777 | 90.9 | - | 5.06 | 4.6 | 82.0 | 69.3 | 6.6 |
| MY[a] | 3981 | 7.3 | - | 0.2 | 0.5 | 23.7 | 40.1 | 1.3 |
| KJ[b] | 2500 | 58.4 | - | 3.3 | 4.3 | 76.4 | 45.6 | 9.4 |
| IJ1[b] | 3300 | 24.4 | 2 | 1.4 | 2.9 | 50.1 | 44.5 | 6.5 |
| IJ2[b] | 3300 | 17.1 | - | 0.6 | 1.2 | 35.1 | 44.5 | 2.6 |
| Fernow1[c] | 1450 | 17.9 | 1 | 1.6 | 0.8 | 9.3 | 23.4 | 3.6 |
| Fernow2[c] | 1450 | 34.3 | 2 | 3.4 | 1.5 | 14.8 | 23.4 | 6.3 |
| Fernow3[c] | 1450 | 60.0 | 3 | 4.2 | 2.4 | 34.5 | 23.4 | 10.3 |
| Uryu[d] | 1170 | 0.7 | - | 0.1 | 0.1 | 1.0 | 18.6 | 0.7 |
| Qingyuan[e] | 709 | 150.0 | 2 | 8.9 | 2.9 | 49.3 | 50.0 | 5.8 |

a: This study
b: Nakagawa et al., 2018; Nakahara et al., 2010
c: Rose et al., 2015
d: Tsunogai et al., 2014
e: Huang et al., 2020
*: N saturation stage estimated in past studies
-: No data

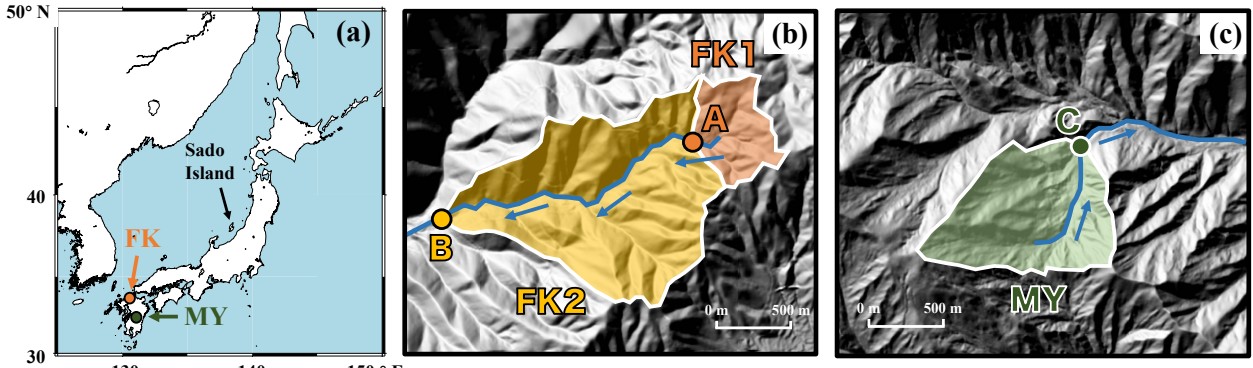

**Figure 1.** A map showing the locations of the study catchments (FK and MY) in Japan
(a), and the maps of FK1, FK2 (b) and MY catchments (c), shown by orange, yellow,
and green areas, respectively, together with the sampling station A, B, and C,
respectively, shown by orange, yellow, and green circles, respectively. The blue arrows
indicate the flow direction of stream water.

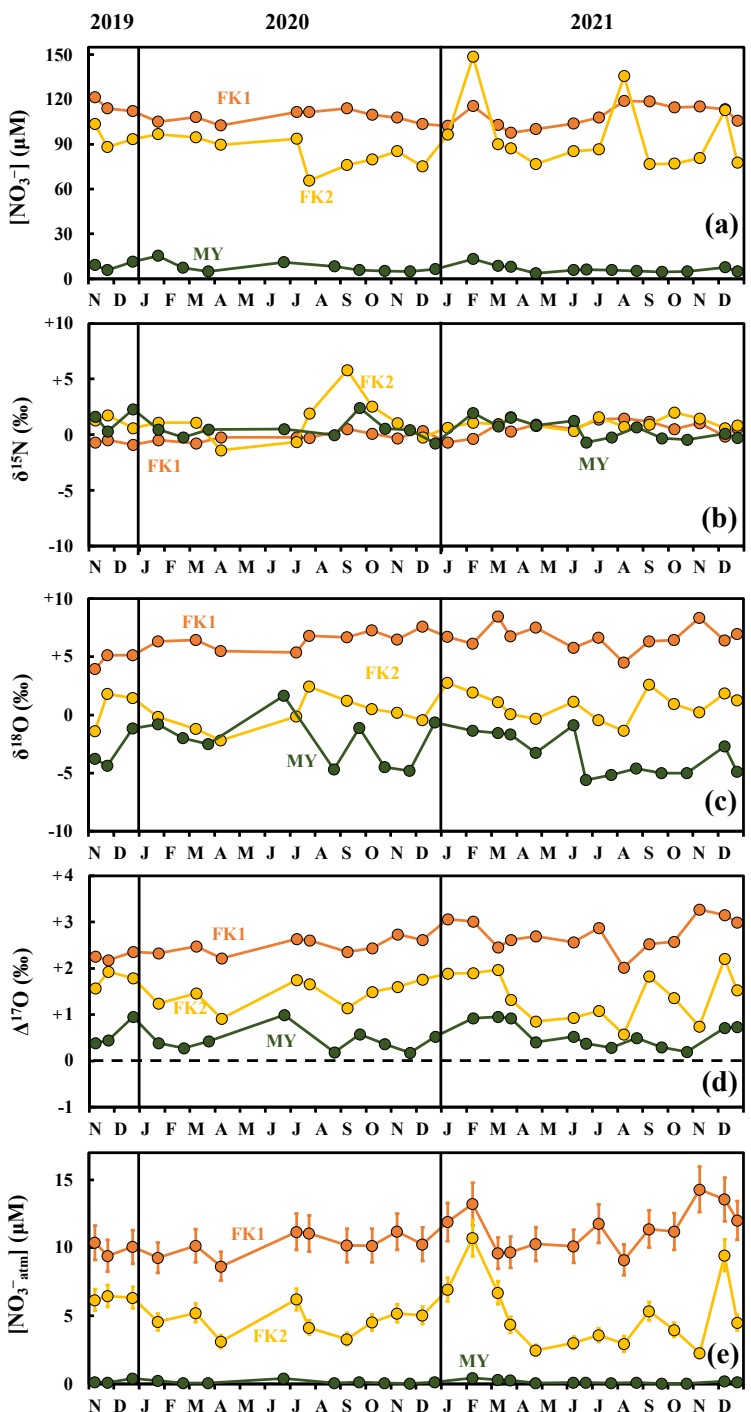

**Figure 2.** Temporal variations in concentrations of stream nitrate (FK1: orange circles;

FK2: yellow circles; MY: green circles) (a), together with those in $\delta^{15}N$ (b), $\delta^{18}O$ (c),

and $\Delta^{17}O$ (d) of nitrate, and the concentration of unprocessed $NO_3^-{}_{atm}$ ($[NO_3^-{}_{atm}]$) (e) in

the stream water of the FK1, FK2, and MY forested catchments. Error bars smaller than

the sizes of the symbols are not presented.

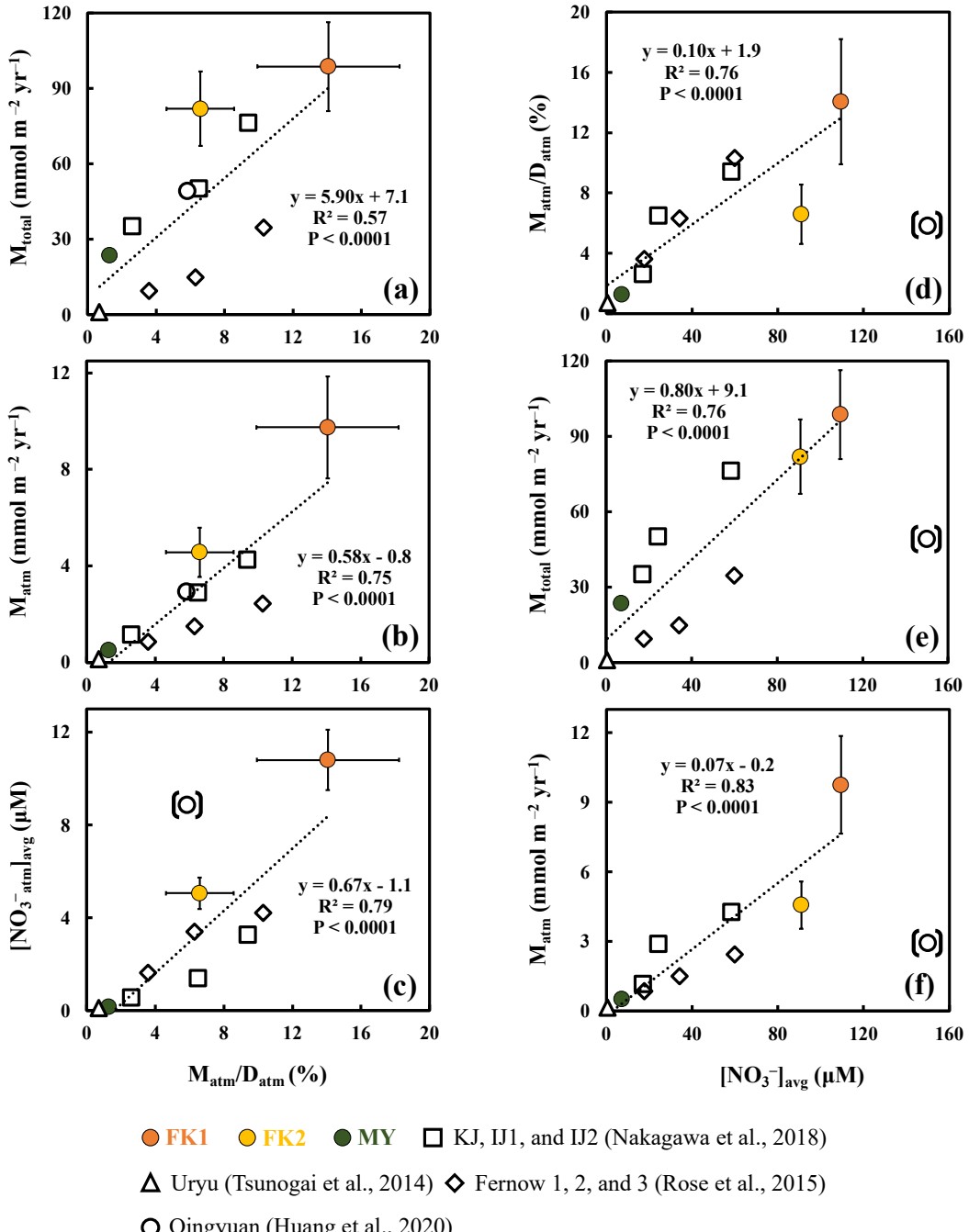

**FK1** (orange circle)  **FK2** (yellow circle)  **MY** (green circle)  □ KJ, IJ1, and IJ2 (Nakagawa et al., 2018)

△ Uryu (Tsunogai et al., 2014)  ◇ Fernow 1, 2, and 3 (Rose et al., 2015)

○ Qingyuan (Huang et al., 2020)

**Figure 3.** Annual export flux of nitrate per unit area ($M_{total}$) plotted as a function of the $M_{atm}/D_{atm}$ ratio in each forested catchment (a); the annual export flux of unprocessed atmospheric nitrate per unit area ($M_{atm}$) plotted as a function of the $M_{atm}/D_{atm}$ ratio (b); the average concentration of $NO_3^-{}_{atm}$ ($[NO_3^-{}_{atm}]_{avg}$) plotted as a function of the $M_{atm}/D_{atm}$ ratio (c); the $M_{atm}/D_{atm}$ ratio plotted as a function of the average concentration

of nitrate ($[NO_3^-]_{avg}$) (d); the $M_{total}$ plotted as a function of $[NO_3^-]_{avg}$ (e); the $M_{atm}$
plotted as a function of $[NO_3^-]_{avg}$ (f) (FK1: orange circles; FK2: yellow circles; MY:
green circles). Those determined for the forested catchments in past studies are plotted
as well (Qingyuan: white circle (Huang et al., 2020); KJ, IJ1, and IJ2: white squares
(Nakagawa et al., 2018); Fernow 1, 2, and 3: white diamonds (Lucy et al., 2015); Uryu:
white triangle (Tsunogai., 2014)). The data obtained in the Qingyuan forested
catchment are shown in parentheses and excluded from the calculation to estimate
correlation coefficients (see text for the reason).

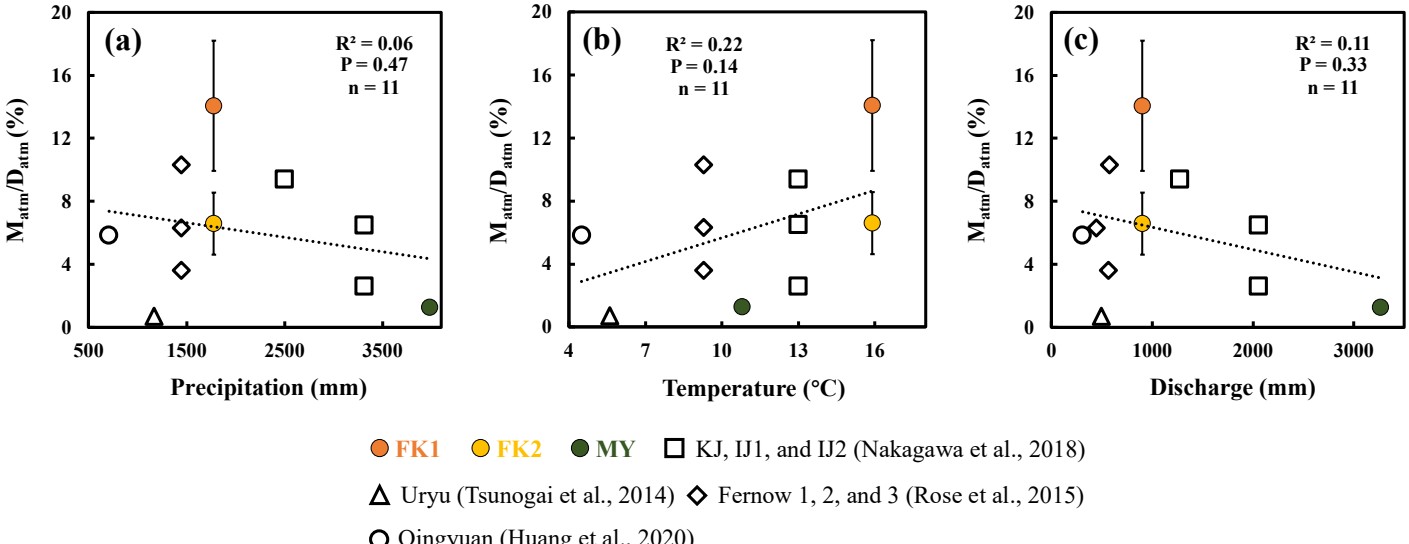

**Figure 4.** the $M_{atm}/D_{atm}$ ratio plotted as a function of the amount of precipitation (a),
the $M_{atm}/D_{atm}$ ratio plotted as a function of the temperature (b), and the $M_{atm}/D_{atm}$ ratio
plotted as a function of flux of stream water (c) (FK1: orange circles; FK2: yellow
circles; MY: green circles). Those determined for the forested catchments in past studies
are plotted as well.

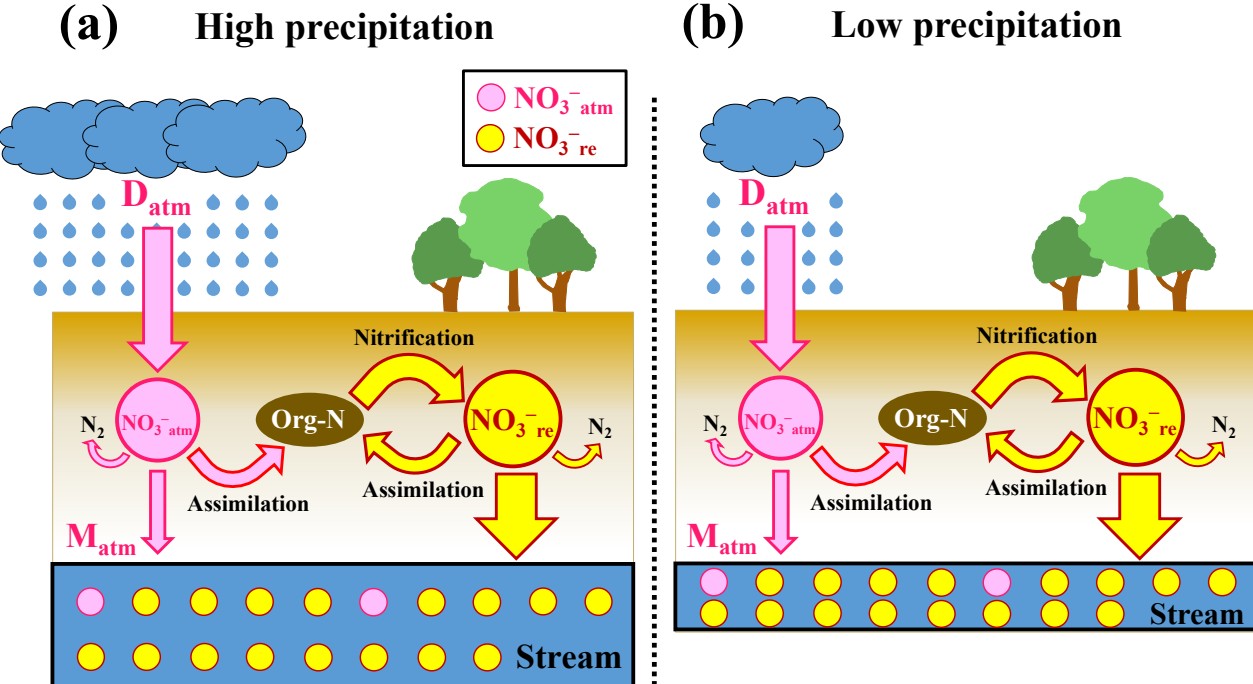

**Figure 5.** Schematic diagram showing the biogeochemical processing of nitrate in forested catchments under high precipitation (a) and low precipitation (b), where $NO_3^-{}_{atm}$ (unprocessed atmospheric nitrate) is represented by pink circles, $NO_3^-{}_{re}$ by yellow circles, the flows of $NO_3^-{}_{atm}$ by pink arrows, and those of $NO_3^-{}_{re}$ (remineralized nitrate) by yellow arrows (modified after Nakagawa., 2018). Although the deposition rates of $NO_3^-{}_{atm}$ ($D_{atm}$) and the biogeochemical reaction rates between (a) and (b) are the same, we can expect high $[NO_3^-]$ in (b). On the other hand, the $M_{atm}/D_{atm}$ ratio between (a) and (b) are the same.