# Peer review of "nitrate from forested catchments under high nitrogen saturation"

_EGUsphere, 2022_

## Author Comment (AC1)

Dear Referee #1

Thank you very much for your valuable comments on our manuscript. We would like to respond to each of your comments and questions one by one.

**We definitely can observed low Matm/Datm ratio if a forest is N limited and almost all precipitation nitrate is biologically processed. However, there are two exceptions. One is high precipitation may cause high Matm/Datm ratio due to limited contact time of precipitation nitrate with soil microbes and roots.**

Thank you for your comment. Our conclusion was derived from FK, MY, and the past data ever reported in forested streams through continuous monitoring on $\Delta^{17}O$ (Table 3 in the Manuscript; Table 1 in this file), where the data of precipitation up to 3837 mm per year, average $[NO_3^-]$, and $M_{atm}/D_{atm}$ ratio were included. While the stream nitrate concentration showed the strong linear relationship ($R^2 = 0.81$; $P < 0.0001$) with the $M_{atm}/D_{atm}$ ratio (Fig. 3d in the Manuscript; Fig. 1a in this file), the amount of precipitation showed no linear relationship ($R^2 = 0.06$; $P = 0.48$) with the $M_{atm}/D_{atm}$ ratio (Fig. 1b). Past studies have used the concentration of stream nitrate as one of the important indexes to evaluate the progress of nitrogen saturation in each forest (Aber, 1992; Huang et al., 2020; Rose et al., 2015; Stoddard, 1994). As a result, we concluded that the $M_{atm}/D_{atm}$ ratio was mainly controlled by the progress of the nitrogen saturation, rather than the amount of the precipitation. We would like to mention this in the revised MS.

**Table 1.** The annual amount of precipitation (P), the average concentration of stream nitrate ($[NO_3^-]_{avg}$), the $M_{atm}/D_{atm}$ ratio, the $\Delta^{17}O$ of atmospheric nitrate, the $\Delta^{17}O$ of stream nitrate, and the gross nitrification rate (GNR) in the FK1, FK2, and MY, along with those in the catchments studied in past studies.

| | Precipitation | $[NO_3^-]_{avg}$ | $M_{atm}/D_{atm}$ | $D_{atm}$ | $\Delta^{17}O(NO_3^-)_{atm}$ | $\Delta^{17}O(NO_3^-)_{stream}$ | GNR |
|---|---|---|---|---|---|---|---|
| | mm | μM | % | mmol m$^{-2}$ yr$^{-1}$ | ‰ | ‰ | mmol m$^{-2}$ yr$^{-1}$ |
| FK1[a] | 1769 | 109.5 | 13.9 | 69.3 | 26.3 | 2.6 | 631.7 |
| FK2[a] | 1769 | 94.2 | 7.9 | 69.3 | 26.3 | 1.7 | 1002.8 |
| My[a] | 3837 | 7.1 | 1.2 | 40.1 | 26.3 | 0.6 | 1718.3 |
| KJ[b] | 2500 | 58.4 | 9.4 | 45.6 | 26.3 | 1.5 | 759.3 |
| IJ1[b] | 3300 | 24.4 | 6.5 | 44.5 | 26.3 | 1.5 | 735.7 |
| IJ2[b] | 3300 | 17.1 | 2.6 | 44.5 | 26.3 | 0.9 | 1332.4 |
| Fellow1[c] | 1450 | 17.9 | 3.6 | 23.4 | 21.3 | 1.9 | 236.6 |
| Fellow2[c] | 1450 | 34.3 | 6.3 | 23.4 | 21.3 | 2.1 | 210.6 |
| Fellow3[c] | 1450 | 60.0 | 10.3 | 23.4 | 21.3 | 1.5 | 311.1 |
| Uryu[d] | 1170 | 0.7 | 0.7 | 18.6 | 26.3 | 8.8 | 36.9 |
| Qingyuan[e] | 709 | 150.0 | 5.8 | 50 | 27.0 | 1.6 | 793.8 |

a: This study
b: Nakagawa et al., 2018
c: Rose et al., 2015
d: Tsunogai et al., 2014
e: Huang et al., 2020

[Figure]

● FK1  ● FK2  ● MY  □ KJ, IJ1, and IJ2 (Nakagawa et al., 2018)  ○ Qingyuan (Huang et al., 2020)
△ Uryu (Tsunogai et al., 2014)  ◇ Fernow 1, 2, and 3 (Rose et al., 2015)

**Figure 1.** the $M_{atm}/D_{atm}$ ratio plotted as a function of the average concentration of nitrate ($[NO_3^-]_{avg}$) (a), the $M_{atm}/D_{atm}$ ratio plotted as a function of the precipitation (b) and the $M_{atm}/D_{atm}$ ratio plotted as a function of the gross nitrification rate (GNR).

**The other is high soil nitrate production (gross nitrification rate), which can dilute of 17O of precipitation nitrate that reachs the soil.**

Thank you for your comment. For the aspect of calculating, the high or low gross nitrification rate (GNR) does not influence the annual export flux of $NO_3^-{}_{atm}$ ($M_{atm}$), and thus the the $M_{atm}/D_{atm}$ ratio. For the aspect of the GNR influence the nitrogen saturation of forest and thus the $M_{atm}/D_{atm}$ ratio, we would like to discuss.

Past studies determined the gross nitrification rate (GNR) in the forested catchments based on the elution flux of unprocessed atmospheric nitrate and remineralized nitrate via stream, determined from the $\Delta^{17}O$ values of $NO_3^-$ in stream water eluted from the catchment, and deposition flux of atmospheric nitrate into the catchment (Riha et al., 2014; Fang et al., 2015; Hattori et al., 2019; Huang et al., 2020).

$$GNR = D_{atm} \times (\Delta^{17}O(NO_3^-)_{atm} - \Delta^{17}O(NO_3^-)_{stream}) / \Delta^{17}O(NO_3^-)_{stream} \qquad (1)$$

where $D_{atm}$ denote the deposition flux of nitrate into the catchments, $\Delta^{17}O(NO_3^-)_{atm}$ and $\Delta^{17}O(NO_3^-)_{stream}$ denote the $\Delta^{17}O$ value of atmospheric nitrate and stream nitrate, respectively.

We compiled all past data ever reported in forested streams through continuous monitoring, where the data of $D_{atm}$, $\Delta^{17}O(NO_3^-)_{atm}$, $\Delta^{17}O(NO_3^-)_{stream}$, GNR, and the $M_{atm}/D_{atm}$ ratio were included (Table 1 in this file). The GNR showed no linear relationship ($R^2 = 0.04$; $P = 0.57$) with the $M_{atm}/D_{atm}$ ratio (Fig. 1c). As a result, the GNR have no influence with the $M_{atm}/D_{atm}$ ratio.

**The streamwater samples for the three forested catchments were collected in 2019 to 2021, while 17O of precipitation nitrate used in the calculation was from the site Sado island in central Japan during 2009 to 2012. So the space and time both were mismatched between stream water sampling sites and precipitation sites. So it is better that authors justified the mismatch. In addition, the average of 17O in precipitation nitrate were used. However, there are a number of studies reporting highly seasonal variation of 17O in precipitation nitrate.**

Thank you for your advice and question. We estimated the uncertainty derived from the difference in the locality as 1 ‰ (Nakagawa et al., 2018). This was based on the standard deviation between the annual average $\Delta^{17}O$ values determined in four different monitoring stations located in the same mid-latitudes, in the past studies such as La Jolla (33° N; Michalski et al., 2003), Princeton (40° N; Kaiser et al., 2007), Rishiri (45° N; Tsunogai et al., 2010), and Sado (38° N; Tsunogai et al., 2016). Besides, we estimated the uncertainty derived from the seasonal difference in the $\Delta^{17}O$ values of atmospheric nitrate as 1.8 ‰, based on the standard deviation of six-month moving averages of atmospheric nitrate determined at the Sado monitoring station. Adding an additional 0.2 ‰ as a margin, we adopted 3 ‰ as the possible error for $\Delta^{17}O$ atm in the streams (we mentioned that in Line 258-261 of manuscript). Additionally, the residence time of groundwater is longer than a few months for most forested catchments in Japan with a humid temperate climate (Takimoto et al., 1994; Kabeya et al., 2007). As a result, seasonal variation of the $\Delta^{17}O$ values of atmospheric

nitrate in the forested catchments in Japan will be buffered by groundwater and the uncertainty of 1.8 ‰ is enough for the seasonal difference in the $\Delta^{17}O$ values of atmospheric nitrate. In addition, Tsunogai et al. (2010) reported the $\Delta^{17}O$ values of atmospheric nitrate in Rishiri as +26.2 ‰ for 2006 to 2007. Tsunogai et al. (2016) reported the the $\Delta^{17}O$ values of atmospheric nitrate in Sado island as +25.5 ‰ for 2009, +27.2 ‰ for 2010 and +25.7 ‰ for 2011. As a result, the temporal variation of the $\Delta^{17}O$ values of atmospheric nitrate can be negligible. We would like to clarify this in the revised MS.

We would like to thank you for the helpful comments and suggestions. We hope that our responses to your comments and questions are satisfactory.

Sincerely,
Weitian Ding
PhD student
Graduate School of Environmental Studies,
Nagoya University
Furo-cho, Chikusa-ku, Nagoya,
464-8601, JAPAN
Phone: +81-70-4436-3157
E-mail: ding.weitian.v2@s.mail.nagoya-u.ac.jp

**Reference**
Aber, J. D.: Nitrogen cycling and nitrogen saturation in temperate forest ecosystems, Trends Ecol. Evol., 7(7), 220–224, doi:10.1016/0169-5347(92)90048-G, 1992.
Fang, Y., Koba, K., Makabe, A., Takahashi, C., Zhu, W., Hayashi, T., Hokari, A. A., Urakawa, R., Bai, E., Houlton, B. Z., Xi, D., Zhang, S., Matsushita, K., Tu, Y., Liu, D., Zhu, F., Wang, Z., Zhou, G., Chen, D., Makita, T., Toda, H., Liu, X., Chen, Q., Zhang, D., Li, Y. and Yoh, M.: Microbial denitrification dominates nitrate losses from forest ecosystems, Proc. Natl. Acad. Sci. U. S. A., 112(5), 1470–1474, doi:10.1073/pnas.1416776112, 2015.
Huang, S., Wang, F., Elliott, E. M., Zhu, F., Zhu, W., Koba, K., Yu, Z., Hobbie, E. A., Michalski, G., Kang, R., Wang, A., Zhu, J., Fu, S. and Fang, Y.: Multiyear Measurements on $\Delta^{17}O$ of Stream Nitrate Indicate High Nitrate Production in a Temperate Forest, Environ. Sci. Technol., 54(7), 4231–4239, doi:10.1021/acs.est.9b07839, 2020.
Hattori, S., Nuñez Palma, Y., Itoh, Y., Kawasaki, M., Fujihara, Y., Takase, K. and Yoshida, N.: Isotopic evidence for seasonality of microbial internal nitrogen cycles in a temperate forested catchment with heavy snowfall, Sci. Total Environ., 690, 290–299, doi:10.1016/j.scitotenv.2019.06.507, 2019.

Kabeya, N., Katsuyama, M., Kawasaki, M., Ohte, N., and Sugi- moto, A.: Estimation of mean residence times of subsurface wa- ters using seasonal variation in deuterium excess in a small head- water catchment in Japan, Hydrol. Process., 21, 308–322, 2007.

Kaiser, J., Hastings, M. G., Houlton, B. Z., Röckmann, T. and Sigman, D. M.: Triple oxygen isotope analysis of nitrate using the denitrifier method and thermal decomposition of N2O, Anal. Chem., 79(2), 599–607, doi:10.1021/ac061022s, 2007.

Michalski, G., Scott, Z., Kabiling, M. and Thiemens, M. H.: First measurements and modeling of $\Delta$17O in atmospheric nitrate, Geophys. Res. Lett., 30(16), 3–6, doi:10.1029/2003GL017015, 2003.

Nakagawa, F., Tsunogai, U., Obata, Y., Ando, K., Yamashita, N., Saito, T., Uchiyama, S., Morohashi, M. and Sase, H.: Export flux of unprocessed atmospheric nitrate from temperate forested catchments: A possible new index for nitrogen saturation, Biogeosciences, 15(22), 7025–7042, doi:10.5194/bg-15-7025-2018, 2018.

Rose, L. A., Elliott, E. M. and Adams, M. B.: Triple Nitrate Isotopes Indicate Differing Nitrate Source Contributions to Streams Across a Nitrogen Saturation Gradient, Ecosystems, 18(7), 1209–1223, doi:10.1007/s10021-015-9891-8, 2015.

Riha, K. M., Michalski, G., Gallo, E. L., Lohse, K. A., Brooks, P. D. and Meixner, T.: High Atmospheric Nitrate Inputs and Nitrogen Turnover in Semi-arid Urban Catchments, Ecosystems, 17(8), 1309–1325, doi:10.1007/s10021-014-9797-x, 2014.

Stoddard, J. L.: Long-Term Changes in Watershed Retention of Nitrogen, , 223–284, doi:10.1021/ba-1994-0237.ch008, 1994.

Tsunogai, U., Komatsu, D. D., Daita, S., Kazemi, G. A., Nakagawa, F., Noguchi, I. and Zhang, J.: Tracing the fate of atmospheric nitrate deposited onto a forest ecosystem in Eastern Asia using $\Delta^{17}$O, Atmos. Chem. Phys., 10(4), 1809–1820, doi:10.5194/acp-10-1809-2010, 2010.

Tsunogai, U., Miyauchi, T., Ohyama, T., Komatsu, D. D., Nakagawa, F., Obata, Y., Sato, K. and Ohizumi, T.: Accurate and precise quantification of atmospheric nitrate in streams draining land of various uses by using triple oxygen isotopes as tracers, Biogeosciences, 13(11), 3441–3459, doi:10.5194/bg-13-3441-2016, 2016.

Tsunogai, U., Komatsu, D. D., Ohyama, T., Suzuki, A., Nakagawa, F., Noguchi, I., Takagi, K. and Nomura, M.: Quantifying the effects of clear-cutting and strip-cutting on nitrate dynamics in a forested watershed using triple oxygen isotopes as tracers, , (1), 5411–5424, doi:10.5194/bg-11-5411-2014, 2014.

Takimoto, H., Tanaka, T., and Horino, H.: Does forest conserve runoff discharge during drought?, Transactions of The Japanese Society of Irrigation, Drainage and Reclamation Engineering, 170, 75–81, https://doi.org/10.11408/jsidre1965.1994.170_75, 1994 (in Japanese with English abstract)

---

## Author Comment (AC2)

Dear Referee #2

Thank you very much for your valuable comments on our manuscript. We would like to respond to each of your comments and questions one by one.

**It is difficult to identify a single driver for the differences in the proportion of atmospheric NO3- export between the two sites given that they differ both in terms of the amount of N deposition and their climate (the low deposition site receives significantly less rainfall and is significantly cooler than the high deposition site; L120-121). Differences in hydrology are not accounted for, but should be (e.g., both surface water – groundwater interactions and slope, both of which could impact N attenuation and the degree of stream water mixing with microbial NO3- sources).**

Thank you for your comment and advice. Our conclusion was derived from FK, MY, and the past data ever reported in forested streams through continuous monitoring on $\Delta^{17}O$, where the data of amount of precipitation, average $[NO_3^-]$, temperature, amount of discharge, and the $M_{atm}/D_{atm}$ ratio were included (Table 1 in this file). While the stream nitrate concentration showed the strong linear relationship ($R^2 = 0.81$; $P < 0.0001$) with the $M_{atm}/D_{atm}$ ratio (Fig. 1a), the amount of precipitation, temperature, and amount of discharge showed no significant relationship with the $M_{atm}/D_{atm}$ ratio ($P > 0.12$; Figs. 1b, 1c, and 1d). Past studies have used the concentration of stream nitrate as one of the important indexes to evaluate the stage of nitrogen saturation in each forest (Aber, 1992; Huang et al., 2020; Rose et al., 2015; Stoddard, 1994). As a result, we concluded the $M_{atm}/D_{atm}$ ratio was mainly controlled by the progress of the nitrogen saturation, rather than the amount of precipitation, temperature, and hydrology. We would like to mention these in the revised MS.

**Table 1.** The annual amount of precipitation (P), the average concentration of stream nitrate ($[NO_3^-]_{avg}$), amount of precipitation, average temperature, amount of discharge, and $M_{atm}/D_{atm}$ ratio in the FK1, FK2, and MY, along with those in the catchments studied in past studies.

| | $[NO_3^-]_{avg}$ | Precipitation | Temperature | Discharge | $M_{atm}/D_{atm}$ |
| --- | --- | --- | --- | --- | --- |
| | µM | mm | °C | mm | % |
| FK1[a] | 109.5 | 1769 | 15.9 | 894 | 13.9 |
| FK2[a] | 94.2 | 1769 | 15.9 | 894 | 7.9 |
| My[a] | 7.1 | 3837 | 10.8 | 3122 | 1.2 |
| KJ[b] | 58.4 | 2500 | 13 | 1276 | 9.4 |
| IJ1[b] | 24.4 | 3300 | 13 | 2057 | 6.5 |
| IJ2[b] | 17.1 | 3300 | 13 | 2057 | 2.6 |
| Fellow1[c] | 17.9 | 1450 | 9.3 | 567 | 3.6 |
| Fellow2[c] | 34.3 | 1450 | 9.3 | 450 | 6.3 |
| Fellow3[c] | 60.0 | 1450 | 9.3 | 578 | 10.3 |
| Uryu[d] | 0.7 | 1170 | 5.6 | 500 | 0.7 |
| Qingyuan[e] | 150.0 | 709 | 4.5 | 309 | 5.8 |

a: This study

b: Nakagawa et al., 2018

c: Rose et al., 2015

d: Tsunogai et al., 2014

e: Huang et al., 2020

[Figure]

Figure 1. the $M_{atm}/D_{atm}$ ratio plotted as a function of the average concentration of nitrate ($[NO_3^-]_{avg}$) (a), the $M_{atm}/D_{atm}$ ratio plotted as a function of the amount of precipitation (b), the $M_{atm}/D_{atm}$ ratio plotted as a function of the temperature, and the $M_{atm}/D_{atm}$ ratio plotted as a function of the amount of discharge.

**These also led to differences in vegetation between the two sites (L114-119).**

Thank you for your comment. By compare the type and the age of plantations in FK1, FK2, and MY catchments, we concluded that the age and type of plantations caused the reduction in N uptake rates and thus increased of the nitrogen saturation and the $M_{atm}/D_{atm}$ ratio in 4.2 section of manuscript.

**The fact that FK has lower concentrations of atmospheric NO3- at the upstream site than the downstream does indicate that there is unaccounted for hydrologic mixing (or loss) occurring along the stream, which could significantly bias M/D estimates based on a single sampling point (as in the MY catchment).**

Thank you for your comment. FK has higher concentrations of atmospheric nitrate at the upstream site than the downstream, insteadly (Table 3 in manuscript). The higher concentrations of atmospheric nitrate (or higher $M_{atm}/D_{atm}$ ratio) in FK1 catchment than FK2 catchment indicated that progress of the nitrogen saturation was heterogeneities, even in a small area (< 100 ha). As a result, we only discussed the $M_{atm}/D_{atm}$ ratio that the area can be covered by the ridgeline and sampling points in MY catchment (43 ha) and other forested catchments.

**The atmospheric deposition info used to calculate M/D (the crux of the study) were collected over 10 years, but these measurements ended prior to the stream water sampling that is the primary data here. This is a major limitation, given how much atmospheric N deposition can vary month to month and year to year. A robust approach to constrain the uncertainty created by relying on this 'mean' data is required.**

Thank you for your comment. Chiwa (2020) reported the bulk deposition rate of atmospheric nitrate ($D_{atm}$) as 4.7 and 3.4 kg ha$^{-1}$ yr$^{-1}$ for 2009 to 2018 in FK and MY catchments, respectively, which all the $D_{atm}$ showed no temporal variation (decreased or increased trend during 2009 to 2018). The standard deviation (SD) and coefficient of variation (CV) were 0.9 kg ha$^{-1}$ yr$^{-1}$ and 16 % for FK catchments, 0.5 kg ha$^{-1}$ yr$^{-1}$ and 15% for MY catchment, respectively. Besides, the residence time of groundwater is longer than a few months for most forested catchments in Japan with a humid temperate climate (Takimoto et al., 1994; Kabeya et al., 2007). Thus, seasonal variation of $D_{atm}$ in the forested catchments in Japan will be buffered by groundwater. In this study, we assumed the uncertainty of the $D_{atm}$ as 20% (large than 16 % and 15 %) in FK and MY catchments, which is enough for the temporal variation in each forested catchment. We would like to mention this in the revised MS.

**Information is also needed on the exact location of the atmospheric sample collection relative to the streamwater collection sites (in particular for helping to assess whether there might be differences in atmospheric inputs at sites FK1 v FK2)**

Thank you for your advice. We would like to add the information in the revised MS. The GPS of the monitored sites of the atmospheric sample deposition were 33.638155, 130.516719 and 32.372358, 131.144182 for FK and MY forested catchments, respectively. The distance between the atmospheric monitored sites and stream sampling points were 3.9, 2.9, and 4.5 km for FK1, FK2, and MY forested catchments (Calculated from google map).

**L4: The abstract should be revised to start with establishing the 'big picture' issue addressed and aim of the study, rather than jumping straight in to site differences.**

Thank you for your advice. We would like to revise as suggested.

**L4-6: Here and elsewhere, I suggest referring to the sites by name rather than using acronyms, as this will make it easier to connect this to other work on the sites and more intuitive to follow within the manuscript.**

Thank you for your advice. We would like to add the name after the acronym in here in revise as suggested.

**L50: This line suggests that groundwater inputs are greater in humid temperate forests than other biomes, which is as far as I know not true.**

Thank you for your advice. We would like to revise this in revised manuscript.

**L66: Word missing after 'recent'**

Thank you for your advice. We would like to revise this in revised manuscript.

**L93-95: How could the validity of the approach be tested with the collected data?**

Past studies have reported that the forested catchments under the nitrogen saturated exported the elevated levels of nitrate, together with the high concentration of nitrate (Aber et al., 1989; Mitchell et al., 1997; Peterjohn et al., 1996). The higher concentration of nitrate and export flux of nitrate ($M_{total}$) in FK catchments compare to the KJ forested catchment, the maximum value of the $M_{atm}/D_{atm}$ ratio before this study, implied progress of nitrogen saturation in FK catchments were sever. The higher $M_{atm}/D_{atm}$ ratio in FK catchments supported the implication.

**Why is there reason to think that this method wouldn't work in catchments with higher rates of N deposition?**

Because concentration of nitrate and export flux of nitrate of FK forested catchments higher than the KJ forested catchment, where the $M_{atm}/D_{atm}$ ratio was the highest prior to this study. While we expected high $M_{atm}/D_{atm}$ ratio in FK forested catchments, we conducted this study to verify this.

**A clear hypothesis about how and why catchment retain v export atmospheric NO3- will be important for setting up a stronger discussion section.**

Thank you for your advice. We would like to add this in revised manuscript.

**L96: Word missing after 'recent'**

Thank you for your advice. We would like to revise this in revised manuscript.

**L105-107: As above, it is not clear how the reliability of the M/D ratio can be evaluated using these methods. What results would show that it's unreliable?**

If the $M_{atm}/D_{atm}$ ratio would be lower in FK catchments than the other low export flux of nitrate ($M_{total}$) catchments, it was difficult to conclude that the $M_{atm}/D_{atm}$ ratio is reliable as an index of nitrogen saturation.

**L126: How were the boundaries between the FK1 and FK2 catchments determined? Fig. 1 indicates that these sites are both located along the same stream in the same catchment.**

We would like to answer this question later.

**L161-163: More information on internal standards needed (number, delta values, etc). Information on calibration for del17O also needed.**

In this study, we used three kinds of the local laboratory nitrate standards, which were named to be GG01 ($\delta^{15}N$ = –3.07 ‰, $\delta^{18}O$ = +1.10 ‰, and $\Delta^{17}O$ = 0 ‰), HDLW02 ($\delta^{15}N$ = +16.11 ‰, $\delta^{18}O$ = +22.20 ‰), and NF ($\Delta^{17}O$ = +19.16 ‰), which the GG01 and the HDLW02 were used to determine the $\delta^{15}N$ and $\delta^{18}O$ of stream nitrate, and the GG01 and the NF was used to determine the $\Delta^{17}O$ of stream nitrate (Tsunogai et al., 2016; Nakagawa et al., 2018; Ding et al., 2022). The oxygen exchange rate between nitrate and water during the chemical conversion was calculated through Eq. (1):
Oxygen exchange rate (%) = $\Delta^{17}O(N_2O)_{NF}$ / $\Delta^{17}O(NO_3^-)_{NF}$     (1)
where the $\Delta^{17}O(N_2O)_{NF}$ denote the $\Delta^{17}O$ value of $N_2O$ that convert from the NF nitrate, the $\Delta^{17}O(NO_3^-)_{NF}$ denote the $\Delta^{17}O$ value of NF nitrate ($\Delta^{17}O$ = +19.16 ‰) (Tsunogai et al., 2016; Nakagawa et al., 2018; Ding et al., 2022). Thank you for your advising. We would like to clarify this in the revised manuscript.

**L226-229: Were climate conditions (rainfall, stream flow, temperature) significantly different between the years where atmospheric N was measured v the years where stream N was measured?**

We could not find significant differences in both rainfall and temperature between 2009-2018 (the years when atmospheric N was measured) and 2019-2021 (the years

when stream N was measured). We compiled the rainfall and temperature during 2009 to 2021 based on the Japan Meteorological Agency at the nearest Fukuoka station (33°34′N, 130°22′E) and Miyazaki station (31°56′N, 131°24′E) (Fig. 2). There are no significant different of rainfall and temperature between 2009-2018 and 2019-2021 (t-test; all the P > 0.21). Because the stream flow was mainly controlled by the rainfall and temperature, we think the stream flow also have no significant different between 2009-2018 and 2019-2021. We would like to use the average value of them during 2009-2021 in the revised manuscript.

[Figure]

**Figure 2.** Temporal variations in the precipitation and temperature during 2009 to 2021 at Fukuoka province (orange) and Miyazaki province (green).

**L234: Is this a reasonable explanation for the two sites? Some geologic / hydrologic information is needed to support this.**

Yes. By using the water balance method (E (mm) = $31.4T_{avg}$ (°C) + 376), Komatsu et al. (2008) estimated the flux of stream water ($F_{stream}$) of three forested catchments in Japan for ten years. They found the estimated year-to-year $F_{stream}$ were well corresponded to year-to-year observed $F_{stream}$ variations in three forested catchments. The estimated errors were less than 6%, and $R^2$ values were higher than 0.91. Thus, the water balance method was reasonable.

**L236: Given how important this value is for estimated M/D (L264), it would be illustrative to calculate stream flow based on a range rather than a single average value.**

Komatsu et al. (2008) proposed the standard error when use the method to estimate the flux of stream water ($F_{stream}$). The standard error (range) was included in the calculated $M_{atm}/D_{atm}$ ratio.

**L273-275: Did rainfall differ between the two stream water sampled years? This would be useful information for helping interpret differences in NO3- over time.**

No. We also could not find significant difference in rainfall of FK and MY catchments between 2009-2018 and 2019-2021 (Fig. 3) (t-test; all the P > 0.16). We would like to use the average value of rainfall during 2009-2021 in the revised manuscript.

[Figure]

**Figure 3.** Temporal variations in the precipitation during 2009 to 2021 at FK catchments (orange) and MY catchment (green).

**L290: Report in more quantitative terms (what is 'little' variation?)**

Thank you for your advice. We would like to add the relationship information of the concentrations of stream nitrate and the time (month), together with the standard deviation (SD) and the coefficient of variation (CV) of them in the revised MS.

**L302-305: Move to Discussion.**

Thank you for your advice. We would like to revise as suggested.

**L325-329: What is the likely source of the 20% discrepancy? Is this due to differences in method (and if so how / what?) or genuine inter-annual differences in either N inputs or N retention? These points should be expanded on here.**

We think the environmental difference of observation site is likely source of the 20% discrepancy. The assumption should be verified by the observation. However, this is not the target in this study.

**L336-343: The collected data would need to be combined with more detailed meteorological information and/or isotopic modelling in order to determine the source of atmospheric N to the two sites. Consequently this explanation for the differences between the two sites is mostly speculation and does not have much baring on the overall aim of the study (to understand forest N saturation**

**dynamics), so I suggest removing altogether or moving to the site description as part of the explanation for the known difference in N deposition rates between the two locations.**

Thank you for your advice. We would like to revise as suggested.

**L349: But how many locations has this been reported for? Given the relatively small dataset shown in Table 3 I wonder how surprising the relatively high M/D ratio is.**

The average $[NO_3^-{}_{atm}]$ of forested stream have reported by many past studies ((Bostic et al., 2021; Bourgeois et al., 2018b, 2018a; Hattori et al., 2019; Huang et al., 2020; Nakagawa et al., 2018; Rose et al., 2015; Sabo et al., 2016; Tsunogai et al., 2014, 2016). However, for calculating the $M_{atm}/D_{atm}$ ratio, not only the average $[NO_3^-{}_{atm}]$ was needed, the $D_{atm}$ (deposition rate of atmospheric nitrate) and the flux of stream water were also needed. Some past studies have not reported the $D_{atm}$ or the flux of stream water. Thus, the number of the forested catchments we compiled in the Table 3 of manuscript were smaller than the number of the forested catchments that reported the average $[NO_3^-{}_{atm}]$ data we listed.

**Is it likely that other sites around the world will have similar (or even higher!) ratios?**

Yes. We expect the $M_{atm}/D_{atm}$ ratios higher than the FK catchments in forested catchments where the progress of nitrogen saturation is more severe than the FK catchments. We would like to conduct the further observations in the future, when the COVID-19 become stable.

**L353: What else besides Datm could cause the high concentration of NO3(atm) in the stream water? Alternative explanations (if they exist) should be discussed.**

We assumed the happening of the snowmelt or storm events could also cause the high concentration of atmospheric nitrate in the stream water, because the happening of them could bring the atmospheric nitrate to the stream water directly. However, the number of happening of snowmelt in the FK and MY forested catchments can be negligible. Besides, the amount of the snowmelt is smaller than the amount of the precipitation significantly. Additionally, in our recent study, we concluded that the storm events have few impacts on the concentration of atmospheric nitrate in the stream (Ding et al., 2022). Thank you for your advice. We would like to add the information as suggested.

**L370-388: Beyond forest N uptake, what could cause catchment retention of N deposition? E.g., retention in soils or groundwater?**

In this study, the retention is included in uptake.

**L415-418: How does this finding compare to other parts of the world where precipitation is low but N deposition is high (e.g., parts of the southwestern US)?**

We compiled all past data ever reported in forested streams through continuous monitoring in Table 3, where the data of average [$NO_3^-$], average [$NO_{3\ atm}^-$], $M_{atm}$, $M_{total}$, $D_{atm}$, and $M_{atm}/D_{atm}$ ratio were included.

**L421-422: The relationship between precipitation and N losses really cannot be evaluated here given that the stream and precipitation data is decoupled (stream data collected after the precipitation sampling was concluded), and that dynamics are consequently evaluated only at a very broad timescale based on mean average annual precipitation and evapotranspiration for the two sites.**

As we already presented, there was no significant difference in precipitation between 2009-2018 and 2019-2021 (t-test; $P > 0.21$) (Fig. 2 in this file). We would like to use the average value of precipitation during 2009-2021 in the revised manuscript. Besides, the uncertainty in $D_{atm}$, uncertainty in stream water flux, and uncertainty in concentration of unprocessed nitrate in the streams were included in the calculated $M_{atm}/D_{atm}$ ratios. Because the $M_{atm}/D_{atm}$ ratios in FK forested catchments were significantly large, even account for the uncertainties, the $M_{atm}/D_{atm}$ ratios can be an index for evaluating nitrogen saturation.

**Fig. 1: This indicates that sites FK1 and FK2 are just two points along the same stream, meaning that they represent the same catchment. Some clarification is needed in the Methods and here to describe the hydrologic connection between the two locations and whether they should be considered upstream/downstream or two different sub-catchment (in which case this map should be updated to clearly show the catchments).**

On 2021/01/15, we estimated the flow rate of stream water ($F_{stream}$) at sampling point of FK1 and FK2 with the value as 0.85 L/s and 4.75 L/s, respectively, by using the salt dilution method (Sappa et al., 2015). According to the mass balance of water, we can estimate the $F_{stream}$ of FK2 catchment as 3.90 L/s. The ration of $F_{stream}$ (FK2) / $F_{stream}$ (FK1) was 4.59. On the other hand, the ration of Area of FK2 (62 ha) / Area of FK1 (14 ha) was 4.43, which the value was comparable with the ration of $F_{stream}$ (FK2) / $F_{stream}$ (FK1). As a result, the increased $F_{stream}$ (FK2) compared to the $F_{stream}$

(FK1) was origin from the FK2 forested catchment, and thus we think FK1 and FK2 are the different forested catchments.

Besides, we would like to update our map as follow as suggested:

[Figure]

In addition, we would like to update our data that relation to the FK2 by using Eq. (2):

$$X_{(FK2)} = (X_{(FK1+FK2)} * F_{stream(FK1+FK2)} - X_{(FK1)} * F_{stream(FK1)}) / F_{stream(FK2)} \quad\quad (2)$$

where the $F_{stream(FK1)}$, $F_{stream(FK2)}$, and $F_{stream(FK1+FK2)}$ denote the flux of stream water of FK1, FK2, and FK1 + FK2, respectively. $X_{(FK1)}$, $X_{(FK2)}$, and $X_{(FK1+FK2)}$ denote the $[NO_3^-]_{avg}$, $[NO_3^-{}_{atm}]_{avg}$, $M_{total}$, $M_{atm}$, or the $M_{atm}/D_{atm}$ ratio of FK1, FK2, and FK1 + FK2, respectively. The values of 0.85 L/s, 3.90 L/s, and 4.75 L/s were used as the $F_{stream(FK1)}$, $F_{stream(FK2)}$, and $F_{stream(FK1+FK2)}$, respectively. Thank you for your advising. We would like to add the new section of 2.7 to clarify these.

**L126: How were the boundaries between the FK1 and FK2 catchments determined? Fig. 1 indicates that these sites are both located along the same stream in the same catchment.**

Firstly, we determined the sampling point in the map by using the GPS data (33.39.31.2689, 130.32.55.0910 for FK1; 33.39.20.9586, 130.32.18.8808 for FK2) (Fig. 4a in this file). Then, we connected the ridge line and the upstream sampling point, which the area (orange) is the FK1 catchment (Fig. 4b in this file). Lastly, by using the same method, the FK2 catchment area was drawn in Fig 4c.

[Figure]

**Figure 4.** The maps showing how we determined the boundary line of the FK1 and FK2 forested catchments.

We would like to thank you for the helpful comments and suggestions. We hope that our responses to your comments and questions are satisfactory.

Sincerely,
Weitian Ding
PhD student
Graduate School of Environmental Studies,
Nagoya University
Furo-cho, Chikusa-ku, Nagoya,
464-8601, JAPAN
Phone: +81-70-4436-3157
E-mail: ding.weitian.v2@s.mail.nagoya-u.ac.jp

**Reference**
Aber, J. D.: Nitrogen cycling and nitrogen saturation in temperate forest ecosystems, Trends Ecol. Evol., 7(7), 220–224, doi:10.1016/0169-5347(92)90048-G, 1992.

Aber, J. D., Nadelhoffer, K. J., Steudler, P. and Melillo, J. M.: Nitrogen Saturation in Northern Forest Ecosystems, Bioscience, 39(6), 378–386, doi:10.2307/1311067, 1989.

Bostic, J. T., Nelson, D. M., Sabo, R. D. and Eshleman, K. N.: Terrestrial Nitrogen Inputs Affect the Export of Unprocessed Atmospheric Nitrate to Surface Waters: Insights from Triple Oxygen Isotopes of Nitrate, Ecosystems, doi:10.1007/s10021-021-00722-9, 2021.

Bourgeois, I., Savarino, J., Némery, J., Caillon, N., Albertin, S., Delbart, F., Voisin, D. and Clément, J. C.: Atmospheric nitrate export in streams along a montane to urban gradient, Sci. Total Environ., 633, 329–340, doi:10.1016/j.scitotenv.2018.03.141, 2018a.

Bourgeois, I., Savarino, J., Caillon, N., Angot, H., Barbero, A., Delbart, F., Voisin, D. and Clément, J. C.: Tracing the Fate of Atmospheric Nitrate in a Subalpine Watershed Using $\Delta^{17}O$, Environ. Sci. Technol., 52(10), 5561–5570, doi:10.1021/acs.est.7b02395, 2018b.

Chiwa, M.: Ten-year determination of atmospheric phosphorus deposition at three forested sites in Japan, Atmos. Environ., 223(May 2019), 1–7, doi:10.1016/j.atmosenv.2019.117247, 2020.

Ding, W., Tsunogai, U., Nakagawa, F., Sambuichi, T., Sase, H., Morohashi, M., and Yotsuyanagi, H.: Tracing the source of nitrate in a forested stream showing elevated concentrations during storm events, Biogeosciences, 19, 3247–3261, https://doi.org/10.5194/bg-19-3247-2022, 2022.

Fang, Y., Koba, K., Makabe, A., Takahashi, C., Zhu, W., Hayashi, T., Hokari, A. A., Urakawa, R., Bai, E., Houlton, B. Z., Xi, D., Zhang, S., Matsushita, K., Tu, Y., Liu, D., Zhu, F., Wang, Z., Zhou, G., Chen, D., Makita, T., Toda, H., Liu, X., Chen, Q.,

Zhang, D., Li, Y. and Yoh, M.: Microbial denitrification dominates nitrate losses from forest ecosystems, Proc. Natl. Acad. Sci. U. S. A., 112(5), 1470–1474, doi:10.1073/pnas.1416776112, 2015.

Huang, S., Wang, F., Elliott, E. M., Zhu, F., Zhu, W., Koba, K., Yu, Z., Hobbie, E. A., Michalski, G., Kang, R., Wang, A., Zhu, J., Fu, S. and Fang, Y.: Multiyear Measurements on $\Delta^{17}O$ of Stream Nitrate Indicate High Nitrate Production in a Temperate Forest, Environ. Sci. Technol., 54(7), 4231–4239, doi:10.1021/acs.est.9b07839, 2020.

Hattori, S., Nuñez Palma, Y., Itoh, Y., Kawasaki, M., Fujihara, Y., Takase, K. and Yoshida, N.: Isotopic evidence for seasonality of microbial internal nitrogen cycles in a temperate forested catchment with heavy snowfall, Sci. Total Environ., 690, 290–299, doi:10.1016/j.scitotenv.2019.06.507, 2019.

Kabeya, N., Katsuyama, M., Kawasaki, M., Ohte, N., and Sugi- moto, A.: Estimation of mean residence times of subsurface wa- ters using seasonal variation in deuterium excess in a small head- water catchment in Japan, Hydrol. Process., 21, 308–322, 2007.

Komatsu, H., Maita, E. and Otsuki, K.: A model to estimate annual forest evapotranspiration in Japan from mean annual temperature, , 330–340, doi:10.1016/j.jhydrol.2007.10.006, 2008b.

Mitchell, M. J., Iwatsubo, G., Ohrui, K. and Nakagawa, Y.: Nitrogen saturation in Japanese forests: An evaluation, For. Ecol. Manage., 97(1), 39–51, doi:10.1016/S0378-1127(97)00047-9, 1997.

Nakagawa, F., Tsunogai, U., Obata, Y., Ando, K., Yamashita, N., Saito, T., Uchiyama, S., Morohashi, M. and Sase, H.: Export flux of unprocessed atmospheric nitrate from temperate forested catchments: A possible new index for nitrogen saturation, Biogeosciences, 15(22), 7025–7042, doi:10.5194/bg-15-7025-2018, 2018.

Peterjohn, W. T., Adams, M. B. and Gilliam, F. S.: Symptoms of nitrogen saturation in two central Appalachian hardwood forest ecosystems, Biogeochemistry, 35(3), 507–522, doi:10.1007/BF02183038, 1996.

Rose, L. A., Elliott, E. M. and Adams, M. B.: Triple Nitrate Isotopes Indicate Differing Nitrate Source Contributions to Streams Across a Nitrogen Saturation Gradient, Ecosystems, 18(7), 1209–1223, doi:10.1007/s10021-015-9891-8, 2015.

Riha, K. M., Michalski, G., Gallo, E. L., Lohse, K. A., Brooks, P. D. and Meixner, T.: High Atmospheric Nitrate Inputs and Nitrogen Turnover in Semi-arid Urban Catchments, Ecosystems, 17(8), 1309–1325, doi:10.1007/s10021-014-9797-x, 2014.

Stoddard, J. L.: Long-Term Changes in Watershed Retention of Nitrogen, , 223–284, doi:10.1021/ba-1994-0237.ch008, 1994.

Sabo, R. D., Nelson, D. M. and Eshleman, K. N.: Episodic, seasonal, and annual export of atmospheric and microbial nitrate from a temperate forest, Geophys. Res. Lett., 43(2), 683–691, doi:10.1002/2015GL066758, 2016.

Tsunogai, U., Miyauchi, T., Ohyama, T., Komatsu, D. D., Nakagawa, F., Obata, Y., Sato, K. and Ohizumi, T.: Accurate and precise quantification of atmospheric nitrate

in streams draining land of various uses by using triple oxygen isotopes as tracers, Biogeosciences, 13(11), 3441–3459, doi:10.5194/bg-13-3441-2016, 2016.

Tsunogai, U., Komatsu, D. D., Ohyama, T., Suzuki, A., Nakagawa, F., Noguchi, I., Takagi, K. and Nomura, M.: Quantifying the effects of clear-cutting and strip-cutting on nitrate dynamics in a forested watershed using triple oxygen isotopes as tracers, , (1), 5411–5424, doi:10.5194/bg-11-5411-2014, 2014.

Takimoto, H., Tanaka, T., and Horino, H.: Does forest conserve runoff discharge during drought?, Transactions of The Japanese Society of Irrigation, Drainage and Reclamation Engineering, 170, 75–81, https://doi.org/10.11408/jsidre1965.1994.170_75, 1994 (in Japanese with English abstract)

---

## Referee Report (RR1)

There are a few outstanding points that were not sufficiently addressed.

1. Reviewer #1 raises a significant point about the fact that high atmospheric N leaching rates can also be caused by hydrology (fast leaching rates) rather than biology (slow production rates). These two scenarios can be thought of as 'kinetic limitation' (not enough time for atmospheric N processing) v 'capacity limitation' (not enough biology to process all received atmospheric N), *sensu* (Lovett and Goodale, 2011). These two competing explanations could not be distinguished based solely on correlations with rainfall amounts. This is because transit time of NO3- through the canopy, soils, and vadose zone will depend on multiple factors, which include rainfall amount as well as soil types, vegetation root structures, and antecedent moisture conditions. The site descriptions, data analysis, and discussion need to be expanded to adequately address the kinetic limitation hypothesis for Matm/Datm dynamics.

2. Based on Fig. 1 supplied in the response to reviewer comments there is a strong inverse relationship between gross nitrification rate and Matm/Datm (i.e., more nitrification means lower export of atmospheric N). It is only the inclusion of literature values that breaks down the relationship. So why is this? A robust discussion that addresses how (or how not) the high gross nitrification rates fit, or don't, the interpretation that Matm/Datm represents differences in catchment N saturation status.

3. More details are needed in the methods section about how uncertainties were incorporated into the findings. The Matm/Datm calculations rely on several assumptions that needed to be made in order to account of lack of data (streamflow) or overlapping measurement periods (atmospheric sampling did not occur on the same years as stream water sampling). There are accordingly a number of significant sources of uncertainty incorporated into the Matm/Datm calculations: the relationship between precip amount and streamflow (which itself incorporates a number of uncertainties: the relationship between temperature and evapotranspiration, potential rate of loss to groundwater), the interannual consistency of 17O of atmospheric nitrate, and the spatial consistency in the amount of rainfall and the 17O of atmospheric nitrate. It is therefore essential to critically evaluate the potential magnitude of impact these assumptions have on the resultant Matm/Datm values. A sensitivity analysis needs to be performed for each parameter, and these ranges need to be clearly represented in the figures, tables, and text.

4. I am still worried about the reliance on, essentially, rainfall and average annual catchment temperature to calculate downstream NO3- discharge. The relationship between rainfall amounts and stream discharge is generally highly complex, and affected by a number of factors such as catchment slope, soils, vegetation, and groundwater connectivity. These factors need to be robustly and quantitatively addressed (i.e., a hydrodynamic model is needed) given how important Fstream is to Matm, and thus the interpretation of systems as N saturated.

5. As a consequence of the above (big) assumption that Fstream = precipitation – evapotransiration, the Matm/Datm ratio is essentially: $\frac{[NO3]stream*(P-E)}{[NO3]bulk*P}$ (ignoring for a moment the calculations around dry and gaseous deposition). This really is then a almost directly a comparison of the concentration of 17O-NO3- measured in stream water over a few years relative to the concentrations of 17O-NO3- measured in the rain over the previous decade, with correction factor for the average annual temperature of the catchment (used to calculate E). Without a more robust approach to uncertainty and stream flow, and a more nuanced discussion of these uncertainties, it is hard to draw any conclusions about ecosystem N saturation from these values. It is also difficult to justify statistical analyses comparing temperature, precipitation, and discharge to Matm/Datm, given that all three parameters are directly used to calculate the ratio (and indeed that temperature and precipitation are themselves used to calculate discharge).

6. I am still very confused about the relationship between FK1 and FK2. Are these in the same catchment or different catchments? Does one flow into the other (referred to as upstream v downstream sites at some points), or do they flow off different sides of a ridge? If the latter, does this affect the amount of precipitation received at both sites? If the former, should these really be considered as independent sites? It also seems the reliance on rain and temperature to determine flow would have a big impact here. Are the streams actually the same size, as would presumably be determined by these calculations?

Lovett, G.M. and Goodale, C.L. (2011) A new conceptual model of nitrogen saturation based on experimental nitrogen addition to an oak forest. Ecosystems 14, 615-631.

---

## Author Response (AR3)

October 26, 2022

Dr. Perran Cook
Editor of Biogeosciences

Title: Stable isotopic evidence for the excess leaching of unprocessed atmospheric nitrate from forested catchments under high nitrogen saturation
Authors: Weitian Ding et al.
MS No.: egusphere-2022-717

Dear Dr. Cook:

Thank you very much for handling our manuscript. We would like to thank the referees as well for the constructive comments on our manuscript. We have carefully studied the comments and revised the manuscript accordingly. We include below point-by-point responses to the comments, and detailed descriptions of the modifications we made to the manuscript. Besides, we also uploaded the revised manuscript in MS Word, in which all the revisions from BGD version were recorded. We hope that with these changes you will find our revised manuscript appropriate for publication in your journal.

Sincerely yours,
Weitian Ding
PhD student
Graduate School of Environmental Studies,
Nagoya University
Furo-cho, Chikusa-ku, Nagoya,
464-8601, JAPAN
Phone: +81-70-4436-3157
E-mail: ding.weitian.v2@s.mail.nagoya-u.ac.jp

**Response to the handing associate editor:**

**>Your abstract starts quite abruptly. Please start with 1-2 sentences of context.**

Thank you for the comment and the advising. "Owing to the elevated loading of nitrogen through atmospheric deposition, some forested ecosystems become nitrogen saturated, from which elevated levels of nitrate are exported." We added the sentence in abstract of the revised MS (P2/L4-6).

**>As noted by reviewer 2, do not use abbreviations especially undefined in the abstract.**

Thank you for the advising. We added the full name of the forested catchments (Kasuya Research forested catchments for FK catchments and Shiiba Research forested catchment for MY catchment) in abstract of the revised MS (P2/L6-7 and P2/L16).

**>Figure 2. As I suggested in your previous manuscript, I am not familiar with the '1000' before the delta values and suggest removing and replace with the 'per mil' symbol at the end.**

Thank you for the suggestion. We revised that in Figure 2 in the revised MS.

**Response to the referee #1:**

**>We definitely can observed low Matm/Datm ratio if a forest is N limited and almost all precipitation nitrate is biologically processed. However, there are two exceptions. One is high precipitation may cause high Matm/Datm ratio due to limited contact time of precipitation nitrate with soil microbes and roots.**

Our conclusion was derived from FK, MY, and the past data ever reported in forested streams through continuous monitoring on $\Delta^{17}O$ (Table 3 in the revised Manuscript), where the data of precipitation up to 3837 mm per year, average $[NO_3^-]$, and $M_{atm}/D_{atm}$ ratio were included. While the stream nitrate concentration showed the strong linear relationship ($R^2 = 0.76$; $P < 0.0001$) with the $M_{atm}/D_{atm}$ ratio (Fig. 3d in the revised MS), the amount of precipitation showed no linear relationship ($R^2 = 0.06$; $P = 0.47$) with the $M_{atm}/D_{atm}$ ratio (Fig. 4a in the revised MS).

Besides, the differences in the number of storm events could affect the $M_{atm}/D_{atm}$ ratio as well, because $NO_3^-{}_{atm}$ could be injected into the stream water directly, along with the storm water (Inamdar and Mitchell, 2006). In recent study, however, we found that the storm events have little impacts on the $M_{atm}/D_{atm}$ ratio, based on monitoring temporal variation of $[NO_3^-{}_{atm}]$ in a stream water during storm events (Ding et al., 2022).

As a result, we concluded that the $M_{atm}/D_{atm}$ ratio was mainly controlled by the progress of nitrogen saturation, rather than the differences in the number of storm events, the amount of precipitation. We mentioned these in the revised MS (P23-24/L450-L468).

Inamdar, S. P. and Mitchell, M. J.: Hydrologic and topographic controls on storm-event exports of dissolved organic carbon (BOC) and nitrate across catchment scales, Water Resour. Res., 42(3), 1–16, doi:10.1029/2005WR004212, 2006.
Ding, W., Tsunogai, U., Nakagawa, F., Sambuichi, T., Sase, H., Morohashi, M., and Yotsuyanagi, H.: Tracing the source of nitrate in a forested stream showing elevated concentrations during storm events, Biogeosciences, 19, 3247–3261, https://doi.org/10.5194/bg-19-3247-2022, 2022.

**>The other is high soil nitrate production (gross nitrification rate), which can dilute of 17O of precipitation nitrate that reachs the soil.**

For the aspect of calculating, the high or low gross nitrification rate (GNR) does not influence the annual export flux of $NO_3^-{}_{atm}$ ($M_{atm}$), and thus the the $M_{atm}/D_{atm}$ ratio. For the aspect of the GNR influence the nitrogen saturation of forest and thus the $M_{atm}/D_{atm}$ ratio, we would like to discuss.

Past studies determined the gross nitrification rate (GNR) in the forested catchments based on the elution flux of unprocessed atmospheric nitrate and

remineralized nitrate via stream, determined from the $\Delta^{17}O$ values of $NO_3^-$ in stream water eluted from the catchment, and deposition flux of atmospheric nitrate into the catchment (Riha et al., 2014; Fang et al., 2015; Hattori et al., 2019; Huang et al., 2020).

$$GNR = D_{atm} \times (\Delta^{17}O(NO_3^-)_{atm} - \Delta^{17}O(NO_3^-)_{stream}) / \Delta^{17}O(NO_3^-)_{stream} \qquad (1)$$

where $D_{atm}$ denote the deposition flux of nitrate into the catchments, $\Delta^{17}O(NO_3^-)_{atm}$ and $\Delta^{17}O(NO_3^-)_{stream}$ denote the $\Delta^{17}O$ value of atmospheric nitrate and stream nitrate, respectively.

The GNR showed no linear relationship ($R^2 = 0.04$; $P = 0.53$; Fig. 1) with the $M_{atm}/D_{atm}$ ratio in all forested catchments. As a result, the GNR have no influence with the $M_{atm}/D_{atm}$ ratio.

[Figure]

**Figure 1.** the $M_{atm}/D_{atm}$ ratio plotted as a function of the gross nitrification rate (GNR).

Fang, Y., Koba, K., Makabe, A., Takahashi, C., Zhu, W., Hayashi, T., Hokari, A. A., Urakawa, R., Bai, E., Houlton, B. Z., Xi, D., Zhang, S., Matsushita, K., Tu, Y., Liu, D., Zhu, F., Wang, Z., Zhou, G., Chen, D., Makita, T., Toda, H., Liu, X., Chen, Q., Zhang, D., Li, Y. and Yoh, M.: Microbial denitrification dominates nitrate losses from forest ecosystems, Proc. Natl. Acad. Sci. U. S. A., 112(5), 1470–1474, doi:10.1073/pnas.1416776112, 2015.

Hattori, S., Nuñez Palma, Y., Itoh, Y., Kawasaki, M., Fujihara, Y., Takase, K. and Yoshida, N.: Isotopic evidence for seasonality of microbial internal nitrogen cycles in a temperate forested catchment with heavy snowfall, Sci. Total Environ., 690, 290–299, doi:10.1016/j.scitotenv.2019.06.507, 2019.

Huang, S., Wang, F., Elliott, E. M., Zhu, F., Zhu, W., Koba, K., Yu, Z., Hobbie, E. A., Michalski, G., Kang, R., Wang, A., Zhu, J., Fu, S. and Fang, Y.: Multiyear Measurements on $\Delta^{17}O$ of Stream Nitrate Indicate High Nitrate Production in a Temperate Forest, Environ. Sci. Technol., 54(7), 4231–4239, doi:10.1021/acs.est.9b07839, 2020.

Riha, K. M., Michalski, G., Gallo, E. L., Lohse, K. A., Brooks, P. D. and Meixner, T.: High Atmospheric Nitrate Inputs and Nitrogen Turnover in Semi-arid Urban

Catchments, Ecosystems, 17(8), 1309–1325, doi:10.1007/s10021-014-9797-x, 2014.

>**The streamwater samples for the three forested catchments were collected in 2019 to 2021, while 17O of precipitation nitrate used in the calculation was from the site Sado island in central Japan during 2009 to 2012. So the space and time both were mismatched between stream water sampling sites and precipitation sites. So it is better that authors justified the mismatch. In addition, the average of 17O in precipitation nitrate were used. However, there are a number of studies reporting highly seasonal variation of 17O in precipitation nitrate.**

We estimated the uncertainty derived from the difference in the locality as 1 ‰ (Nakagawa et al., 2018). This was based on the standard deviation between the annual average $\Delta^{17}O$ values determined in four different monitoring stations located in the same mid-latitudes, in the past studies such as La Jolla (33° N; Michalski et al., 2003), Princeton (40° N; Kaiser et al., 2007), Rishiri (45° N; Tsunogai et al., 2010), and Sado (38° N; Tsunogai et al., 2016). Besides, we estimated the uncertainty derived from the seasonal difference in the $\Delta^{17}O$ values of atmospheric nitrate as 1.8 ‰, based on the standard deviation of six-month moving averages of atmospheric nitrate determined at the Sado monitoring station. Adding an additional 0.2 ‰ as a margin, we adopted 3 ‰ as the possible error for $\Delta^{17}O$ atm in the streams (we mentioned that in Line 258-261 of manuscript). Additionally, the residence time of groundwater is longer than a few months for most forested catchments in Japan with a humid temperate climate (Takimoto et al., 1994; Kabeya et al., 2007). As a result, seasonal variation of the $\Delta^{17}O$ values of atmospheric nitrate in the forested catchments in Japan will be buffered by groundwater and the uncertainty of 1.8 ‰ is enough for the seasonal difference in the $\Delta^{17}O$ values of atmospheric nitrate. In addition, Tsunogai et al. (2010) reported the $\Delta^{17}O$ values of atmospheric nitrate in Rishiri as +26.2 ‰ for 2006 to 2007. Tsunogai et al. (2016) reported the the $\Delta^{17}O$ values of atmospheric nitrate in Sado island as +25.5 ‰ for 2009, +27.2 ‰ for 2010 and +25.7 ‰ for 2011. As a result, the temporal variation of the $\Delta^{17}O$ values of atmospheric nitrate can be negligible.

Kabeya, N., Katsuyama, M., Kawasaki, M., Ohte, N., and Sugi- moto, A.: Estimation of mean residence times of subsurface wa- ters using seasonal variation in deuterium excess in a small head- water catchment in Japan, Hydrol. Process., 21, 308–322, 2007.
Kaiser, J., Hastings, M. G., Houlton, B. Z., Röckmann, T. and Sigman, D. M.: Triple oxygen isotope analysis of nitrate using the denitrifier method and thermal decomposition of N2O, Anal. Chem., 79(2), 599–607, doi:10.1021/ac061022s, 2007.
Michalski, G., Scott, Z., Kabiling, M. and Thiemens, M. H.: First measurements and modeling of $\Delta$17O in atmospheric nitrate, Geophys. Res. Lett., 30(16), 3–6, doi:10.1029/2003GL017015, 2003.

Nakagawa, F., Tsunogai, U., Obata, Y., Ando, K., Yamashita, N., Saito, T., Uchiyama, S., Morohashi, M. and Sase, H.: Export flux of unprocessed atmospheric nitrate from temperate forested catchments: A possible new index for nitrogen saturation, Biogeosciences, 15(22), 7025–7042, doi:10.5194/bg-15-7025-2018, 2018.

Tsunogai, U., Komatsu, D. D., Daita, S., Kazemi, G. A., Nakagawa, F., Noguchi, I. and Zhang, J.: Tracing the fate of atmospheric nitrate deposited onto a forest ecosystem in Eastern Asia using $\Delta^{17}O$, Atmos. Chem. Phys., 10(4), 1809–1820, doi:10.5194/acp-10-1809-2010, 2010.

Tsunogai, U., Miyauchi, T., Ohyama, T., Komatsu, D. D., Nakagawa, F., Obata, Y., Sato, K. and Ohizumi, T.: Accurate and precise quantification of atmospheric nitrate in streams draining land of various uses by using triple oxygen isotopes as tracers, Biogeosciences, 13(11), 3441–3459, doi:10.5194/bg-13-3441-2016, 2016.

Takimoto, H., Tanaka, T., and Horino, H.: Does forest conserve runoff discharge during drought?, Transactions of The Japanese Society of Irrigation, Drainage and Reclamation Engineering, 170, 75–81, https://doi.org/10.11408/jsidre1965.1994.170_75, 1994 (in Japanese with English abstract)

**Response to the referee #2:**

**>It is difficult to identify a single driver for the differences in the proportion of atmospheric NO3- export between the two sites given that they differ both in terms of the amount of N deposition and their climate (the low deposition site receives significantly less rainfall and is significantly cooler than the high deposition site; L120-121). Differences in hydrology are not accounted for, but should be (e.g., both surface water – groundwater interactions and slope, both of which could impact N attenuation and the degree of stream water mixing with microbial NO3- sources).**

Our conclusion was derived from FK, MY, and the past data ever reported in forested streams through continuous monitoring on $\Delta^{17}O$, where the data of amount of precipitation, average $[NO_3^-]$, temperature, amount of discharge, and the $M_{atm}/D_{atm}$ ratio were included (Table S1 in the revised supplement). While the stream nitrate concentration showed the strong linear relationship ($R^2 = 0.76$; $P < 0.0001$) with the $M_{atm}/D_{atm}$ ratio (Fig. 3d in the revised MS), the amount of precipitation, temperature, and amount of discharge showed no significant relationship with the $M_{atm}/D_{atm}$ ratio ($P > 0.14$; Fig. 4 in the revised MS). As a result, we concluded the $M_{atm}/D_{atm}$ ratio was mainly controlled by the progress of the nitrogen saturation, rather than the amount of precipitation, temperature, and hydrology. We mentioned these in the revised MS (P23-24/L459-L468).

**>These also led to differences in vegetation between the two sites (L114-119).**

By compare the type and the age of plantations in FK1, FK2, and MY catchments, we concluded that the age and type of plantations caused the reduction in N uptake rates and thus increased of the nitrogen saturation and the $M_{atm}/D_{atm}$ ratio in 4.2 section of MS.

**>The fact that FK has lower concentrations of atmospheric NO3- at the upstream site than the downstream does indicate that there is unaccounted for hydrologic mixing (or loss) occurring along the stream, which could significantly bias M/D estimates based on a single sampling point (as in the MY catchment).**

FK catchments have higher concentrations of atmospheric nitrate at the upstream site than the downstream, insteadly (Table 3 in MS). The higher concentrations of atmospheric nitrate (or higher $M_{atm}/D_{atm}$ ratio) in FK1 catchment than FK2 catchment indicated that progress of the nitrogen saturation was heterogeneities, even in a small area (< 100 ha). As a result, we only discussed the $M_{atm}/D_{atm}$ ratio that the area can be covered by the ridgeline and sampling points in MY catchment (43 ha) and other forested catchments.

**>The atmospheric deposition info used to calculate M/D (the crux of the study) were collected over 10 years, but these measurements ended prior to the stream water sampling that is the primary data here. This is a major limitation, given how much atmospheric N deposition can vary month to month and year to year. A robust approach to constrain the uncertainty created by relying on this 'mean' data is required.**

Chiwa (2020) reported the bulk deposition rate of atmospheric nitrate ($D_{atm}$) as 4.7 and 3.4 kg ha$^{-1}$ yr$^{-1}$ for 2009 to 2018 in FK and MY catchments, respectively, which all the $D_{atm}$ showed no temporal variation (decreased or increased trend during 2009 to 2018). The standard deviation (SD) and coefficient of variation (CV) were 0.9 kg ha$^{-1}$ yr$^{-1}$ and 16 % for FK catchments, 0.5 kg ha$^{-1}$ yr$^{-1}$ and 15% for MY catchment, respectively. Besides, the residence time of groundwater is longer than a few months for most forested catchments in Japan with a humid temperate climate (Takimoto et al., 1994; Kabeya et al., 2007). Thus, seasonal variation of $D_{atm}$ in the forested catchments in Japan will be buffered by groundwater. In this study, we assumed the uncertainty of the $D_{atm}$ as 20% (large than 16 % and 15 %) in FK and MY catchments, which is enough for the temporal variation in each forested catchment.

Kabeya, N., Katsuyama, M., Kawasaki, M., Ohte, N., and Sugi- moto, A.: Estimation of mean residence times of subsurface wa- ters using seasonal variation in deuterium excess in a small head- water catchment in Japan, Hydrol. Process., 21, 308–322, 2007.
Takimoto, H., Tanaka, T., and Horino, H.: Does forest conserve runoff discharge during drought?, Transactions of The Japanese Society of Irrigation, Drainage and Reclamation Engineering, 170, 75–81, https://doi.org/10.11408/jsidre1965.1994.170_75, 1994 (in Japanese with English abstract)

**>Information is also needed on the exact location of the atmospheric sample collection relative to the streamwater collection sites (in particular for helping to assess whether there might be differences in atmospheric inputs at sites FK1 v FK2)**

The distances between the monitoring sites of bulk deposition in the FK1, FK2, and MY forested catchments and the stations of stream water sampling were 3.9, 2.9, and 4.5 km, respectively (Calculated from google map). We mentioned this in the revised MS (P11/L209-211).

**>L4: The abstract should be revised to start with establishing the 'big picture' issue addressed and aim of the study, rather than jumping straight in to site differences.**

"Owing to the elevated loading of nitrogen through atmospheric deposition, some forested ecosystems become nitrogen saturated, from which elevated levels of nitrate are exported." We added the sentence in abstract of the revised MS (P2/L4-6).

**>L4-6: Here and elsewhere, I suggest referring to the sites by name rather than using acronyms, as this will make it easier to connect this to other work on the sites and more intuitive to follow within the manuscript.**

We added the full name of the forested catchments (Kasuya Research forested catchments for FK catchments and Shiiba Research forested catchment for MY catchment) in abstract of the revised MS (P2/L6-7 and P2/L16).

**>L50: This line suggests that groundwater inputs are greater in humid temperate forests than other biomes, which is as far as I know not true.**

We revised this in the revised MS (P4/L53-56).
"seasonal variation of soil nitrate can be buffered by groundwater with long residence time, so that the seasonal variation is unclear in stream nitrate concentration in Japan, even in normal forests under the nitrogen saturation stage of 0 (Mitchell et al., 1997)"

**>L66: Word missing after 'recent'**

We revised this in the revised MS (P5/L70).

**>L93-95: How could the validity of the approach be tested with the collected data?**

Past studies have reported that the forested catchments under the nitrogen saturated exported the elevated levels of nitrate, together with the high concentration of nitrate (Aber et al., 1989; Mitchell et al., 1997; Peterjohn et al., 1996). The higher concentration of nitrate and export flux of nitrate ($M_{total}$) in FK catchments compare to the KJ forested catchment, the maximum value of the $M_{atm}/D_{atm}$ ratio before this study, implied progress of nitrogen saturation in FK catchments were sever. The higher $M_{atm}/D_{atm}$ ratio in FK catchments supported the implication.

Aber, J. D.: Nitrogen cycling and nitrogen saturation in temperate forest ecosystems, Trends Ecol. Evol., 7(7), 220–224, doi:10.1016/0169-5347(92)90048-G, 1992.
Aber, J. D., Nadelhoffer, K. J., Steudler, P. and Melillo, J. M.: Nitrogen Saturation in

Northern Forest Ecosystems, Bioscience, 39(6), 378–386, doi:10.2307/1311067, 1989.

Mitchell, M. J., Iwatsubo, G., Ohrui, K. and Nakagawa, Y.: Nitrogen saturation in Japanese forests: An evaluation, For. Ecol. Manage., 97(1), 39–51, doi:10.1016/S0378-1127(97)00047-9, 1997.

Peterjohn, W. T., Adams, M. B. and Gilliam, F. S.: Symptoms of nitrogen saturation in two central Appalachian hardwood forest ecosystems, Biogeochemistry, 35(3), 507–522, doi:10.1007/BF02183038, 1996.

**>Why is there reason to think that this method wouldn't work in catchments with higher rates of N deposition?**

Because concentration of nitrate and export flux of nitrate of FK forested catchments higher than the KJ forested catchment, where the $M_{atm}/D_{atm}$ ratio was the highest prior to this study. While we expected high $M_{atm}/D_{atm}$ ratio in FK forested catchments, we conducted this study to verify this.

**>A clear hypothesis about how and why catchment retain v export atmospheric NO3- will be important for setting up a stronger discussion section.**

We added this in revised manuscript (P6/L98-100).
"Whether the index of the $M_{atm}/D_{atm}$ ratio can be applied to forested catchments, where the leaching of stream nitrate is much higher than the KJ forested catchment, remained unclarified. Besides, the advantages of the $M_{atm}/D_{atm}$ ratio within the past indexes of nitrogen saturation have not been discussed."

**>L96: Word missing after 'recent'**

We revised this in the revised MS (P6/L101).

**>L105-107: As above, it is not clear how the reliability of the M/D ratio can be evaluated using these methods. What results would show that it's unreliable?**

If the $M_{atm}/D_{atm}$ ratio would be lower in FK catchments than the other low export flux of nitrate ($M_{total}$) catchments, it was difficult to conclude that the $M_{atm}/D_{atm}$ ratio is reliable as an index of nitrogen saturation.

**>L161-163: More information on internal standards needed (number, delta values, etc). Information on calibration for del17O also needed.**

In this study, we used three kinds of the local laboratory nitrate standards, which were named to be GG01 ($\delta^{15}N = –3.07$ ‰, $\delta^{18}O = +1.10$ ‰, and $\Delta^{17}O = 0$ ‰),

HDLW02 ($\delta^{15}N$ = +8.94 ‰, $\delta^{18}O$ = +24.07 ‰), and NF ($\Delta^{17}O$ = +19.16 ‰), which the GG01 and the HDLW02 were used to determine the $\delta^{15}N$ and $\delta^{18}O$ of stream nitrate, and the GG01 and the NF was used to determine the $\Delta^{17}O$ of stream nitrate. The oxygen exchange rate between nitrate and water during the chemical conversion was calculated through Eq. (1):

Oxygen exchange rate (%) = $\Delta^{17}O(N_2O)_{NF}$ / $\Delta^{17}O(NO_3^-)_{NF}$ (1)

where the $\Delta^{17}O(N_2O)_{NF}$ denote the $\Delta^{17}O$ value of $N_2O$ that convert from the NF nitrate, the $\Delta^{17}O(NO_3^-)_{NF}$ denote the $\Delta^{17}O$ value of NF nitrate ($\Delta^{17}O$ = +19.16 ‰). We mentioned these in the revised MS (P9-10/L163-174).

**>L226-229: Were climate conditions (rainfall, stream flow, temperature) significantly different between the years where atmospheric N was measured v the years where stream N was measured?**

We could not find significant differences in both rainfall and temperature between 2009-2018 (the years when atmospheric N was measured) and 2019-2021 (the years when stream N was measured). We compiled the rainfall and temperature during 2009 to 2021 based on the Japan Meteorological Agency at the nearest Fukuoka station (33°34′N, 130°22′E) and Miyazaki station (31°56′N, 131°24′E) (Fig. 2). There are no significant different of rainfall and temperature between 2009-2018 and 2019-2021 (t-test; all the P > 0.21). Because the stream flow was mainly controlled by the rainfall and temperature, we think the stream flow also have no significant different between 2009-2018 and 2019-2021. We used the average value of them during 2009-2021 in the revised manuscript (P13/L240-241).

[Figure]

**Figure 2.** Temporal variations in the precipitation and temperature during 2009 to 2021 at Fukuoka province (orange) and Miyazaki province (green).

**>L234: Is this a reasonable explanation for the two sites? Some geologic / hydrologic information is needed to support this.**

Yes. By using the water balance method (E (mm) = $31.4T_{avg}$ (°C) + 376), Komatsu et al. (2008) estimated the flux of stream water ($F_{stream}$) of three forested catchments in Japan for ten years. They found the estimated year-to-year $F_{stream}$ were well corresponded to year-to-year observed $F_{stream}$ variations in three forested catchments. The estimated errors were less than 6%, and $R^2$ values were higher than 0.91. Thus, the water balance method was reasonable.

**>L236: Given how important this value is for estimated M/D (L264), it would be illustrative to calculate stream flow based on a range rather than a single average value.**

Komatsu et al. (2008) proposed the standard error when use the method to estimate the flux of stream water ($F_{stream}$). The standard error (range) was included in the calculated $M_{atm}/D_{atm}$ ratio.

**>L273-275: Did rainfall differ between the two stream water sampled years? This would be useful information for helping interpret differences in NO3- over time.**

No. We also could not find significant difference in rainfall of FK and MY catchments between 2009-2018 and 2019-2021 (Fig. 3) (t-test; all the P > 0.16). We used the average value of rainfall during 2009-2021 in the revised MS (P16/L314-316).

[Figure]

**Figure 3.** Temporal variations in the precipitation during 2009 to 2021 at FK catchments (orange) and MY catchment (green).

**>L290: Report in more quantitative terms (what is 'little' variation?)**

We added the relationship information of the concentrations of stream nitrate and the time (month), together with the standard deviation (SD) and the coefficient of

variation (CV) of them in the revised MS (P17/L334-336). We also revised this in the revised MS (P17/L336-337).

"All catchments showed no clear seasonal variation during the observation periods."

**>L302-305: Move to Discussion.**

We revised this as suggestion (P17/L388-391).

**>L325-329: What is the likely source of the 20% discrepancy? Is this due to differences in method (and if so how / what?) or genuine inter-annual differences in either N inputs or N retention? These points should be expanded on here.**

We think the environmental difference of observation site is likely source of the 20% discrepancy. The assumption should be verified by the observation. However, this is not the target in this study.

**>L336-343: The collected data would need to be combined with more detailed meteorological information and/or isotopic modelling in order to determine the source of atmospheric N to the two sites. Consequently this explanation for the differences between the two sites is mostly speculation and does not have much baring on the overall aim of the study (to understand forest N saturation dynamics), so I suggest removing altogether or moving to the site description as part of the explanation for the known difference in N deposition rates between the two locations.**

We removed the sentence of "As a result, the local emission in the Fukuoka metropolitan area should be responsible for the high $D_{atm}$ at the FK catchments" in the revised MS.

**>L349: But how many locations has this been reported for? Given the relatively small dataset shown in Table 3 I wonder how surprising the relatively high M/D ratio is.**

The average [$NO_3^-{}_{atm}$] of forested stream have reported by many past studies ((Bostic et al., 2021; Bourgeois et al., 2018b, 2018a; Hattori et al., 2019; Huang et al., 2020; Nakagawa et al., 2018; Rose et al., 2015; Sabo et al., 2016; Tsunogai et al., 2014, 2016). However, for calculating the $M_{atm}/D_{atm}$ ratio, not only the average [$NO_3^-{}_{atm}$] was needed, the $D_{atm}$ (deposition rate of atmospheric nitrate) and the flux of stream water were also needed. Some past studies have not reported the $D_{atm}$ or the flux of stream water. Thus, the number of the forested catchments we compiled in the

Table 3 of manuscript were smaller than the number of the forested catchments that reported the average [$NO_3^-{}_{atm}$] data we listed.

**>Is it likely that other sites around the world will have similar (or even higher!) ratios?**

Yes. We expect the $M_{atm}/D_{atm}$ ratios higher than the FK catchments in forested catchments where the progress of nitrogen saturation is more severe than the FK catchments. We would like to conduct the further observations in the future, when the COVID-19 become stable.

**>L353: What else besides Datm could cause the high concentration of NO3(atm) in the stream water? Alternative explanations (if they exist) should be discussed.**

We assumed the happening of the storm or snowmelt could also cause the high concentration of atmospheric nitrate in the stream water, because $NO_3^-{}_{atm}$ could be injected into the stream water directly, along with the storm / snowmelt water (Tsunogai et al., 2014; Ding et al., 2022; Inamdar and Mitchell, 2006). In recent study, however, we found that the storm events have little impacts on the $M_{atm}/D_{atm}$ ratio, based on monitoring temporal variation of [$NO_3^-{}_{atm}$] in a stream water during storm events (Ding et al., 2022). Besides, the number of happening of snowmelt in the FK and MY forested catchments can be negligible. In addition, the amount of the snowmelt is smaller than the amount of the precipitation significantly. We added the information as suggested in the revised MS (P23/L450-458).

Besides, the only concern on using the $M_{atm}/D_{atm}$ ratio as the index of nitrogen saturation is the impact of the differences in the residence time of water in each catchment. The residence time of water varies from 1 month to more than 1 year in forested catchments (Asano et al., 2002; Farrick and Branfireun, 2015; Kabeya et al., 2008; Rodgers et al., 2005; Soulsby et al., 2006; Tetzlaff et al., 2007). The $M_{atm}/D_{atm}$ ratio could be higher in catchments with shorter residence time of water. We would like to clarify this in future studies by adding much more data of stream nitrate eluted from various forested catchments. We mentioned this in the revised MS (P25/L492-499).

Asano, Y., Uchida, T. and Ohte, N.: Residence times and flow paths of water in steep unchannelled catchments, Tanakami, Japan, J. Hydrol., 261(1–4), 173–192, doi:10.1016/S0022-1694(02)00005-7, 2002.

Ding, W., Tsunogai, U., Nakagawa, F., Sambuichi, T., Sase, H., Morohashi, M., and Yotsuyanagi, H.: Tracing the source of nitrate in a forested stream showing elevated concentrations during storm events, Biogeosciences, 19, 3247–3261, https://doi.org/10.5194/bg-19-3247-2022, 2022.

Farrick, K. K. and Branfireun, B. A.: Flowpaths, source water contributions and water

residence times in a Mexican tropical dry forest catchment, J. Hydrol., 529, 854–865, doi:10.1016/j.jhydrol.2015.08.059, 2015.

Inamdar, S. P. and Mitchell, M. J.: Hydrologic and topographic controls on storm-event exports of dissolved organic carbon (BOC) and nitrate across catchment scales, Water Resour. Res., 42(3), 1–16, doi:10.1029/2005WR004212, 2006.

Kabeya, N., Shimizu, A., Nobuhiro, T. and Tamai, K.: Preliminary study of flow regimes and stream water residence times in multi-scale forested watersheds of central Cambodia, Paddy Water Environ., 6(1), 25–35, doi:10.1007/s10333-008-0104-3, 2008.

Rodgers, P., Soulsby, C., Waldron, S. and Tetzlaff, D.: Using stable isotope tracers to identify hydrological flow paths, residence times and landscape controls in a mesoscale catchment, Hydrol. Earth Syst. Sci. Discuss., 9, 139–155, 2005.

Soulsby, C., Tetzlaff, D., Rodgers, P., Dunn, S. and Waldron, S.: Runoff processes, stream water residence times and controlling landscape characteristics in a mesoscale catchment: An initial evaluation, J. Hydrol., 325(1–4), 197–221, doi:10.1016/j.jhydrol.2005.10.024, 2006.

Tamai, K.: Preliminary study of flow regimes and stream water residence times in multi-scale forested watersheds of central Cambodia, Paddy Water Environ., 6(1), 25–35, doi:10.1007/s10333-008-0104-3, 2008.

Tetzlaff, D., Malcolm, I. A. and Soulsby, C.: Influence of forestry, environmental change and climatic variability on the hydrology, hydrochemistry and residence times of upland catchments, J. Hydrol., 346(3–4), 93–111, doi:10.1016/j.jhydrol.2007.08.016, 2007.

Tsunogai, U., Komatsu, D. D., Ohyama, T., Suzuki, A., Nakagawa, F., Noguchi, I., Takagi, K. and Nomura, M.: Quantifying the effects of clear-cutting and strip-cutting on nitrate dynamics in a forested watershed using triple oxygen isotopes as tracers, , (1), 5411–5424, doi:10.5194/bg-11-5411-2014, 2014.

**>L370-388: Beyond forest N uptake, what could cause catchment retention of N deposition? E.g., retention in soils or groundwater?**

In this study, the retention is included in uptake.

**>L415-418: How does this finding compare to other parts of the world where precipitation is low but N deposition is high (e.g., parts of the southwestern US)?**

We compiled all past data ever reported in forested streams through continuous monitoring in Table 3, where the data of average $[NO_3^-]$, average $[NO_{3\ atm}^-]$, $M_{atm}$, $M_{total}$, $D_{atm}$, and $M_{atm}/D_{atm}$ ratio were included.

**>L421-422: The relationship between precipitation and N losses really cannot be evaluated here given that the stream and precipitation data is decoupled (stream**

**data collected after the precipitation sampling was concluded), and that dynamics are consequently evaluated only at a very broad timescale based on mean average annual precipitation and evapotranspiration for the two sites.**

There was no significant difference in precipitation between 2009-2018 and 2019-2021 (t-test; P > 0.16) (Fig. 3). We used the average value of precipitation during 2009-2021 in the revised MS. Besides, the uncertainty in $D_{atm}$, uncertainty in stream water flux, and uncertainty in concentration of unprocessed nitrate in the streams were included in the calculated $M_{atm}/D_{atm}$ ratios. Because the $M_{atm}/D_{atm}$ ratios in FK1 forested catchment was significantly large, even account for the uncertainties, the $M_{atm}/D_{atm}$ ratios can be an index for evaluating nitrogen saturation.

**>Fig. 1: This indicates that sites FK1 and FK2 are just two points along the same stream, meaning that they represent the same catchment. Some clarification is needed in the Methods and here to describe the hydrologic connection between the two locations and whether they should be considered upstream/downstream or two different sub-catchment (in which case this map should be updated to clearly show the catchments).**

We updated the map as follow as suggested in the revised MS (Fig.1 in the revised MS):

[Figure]

In addition, we added a new section of 2.7 as fellow to update the data that relation to FK2 catchment (P15-16/L282-310).

2.7 Concentration and isotopic compositions of stream nitrate eluted only from the FK2 catchment

The concentration and isotopic compositions ($\delta^{15}N$, $\delta^{18}O$, and $\Delta^{17}O$) of stream nitrate determined at the station B were the mixture of those eluted from FK1 and FK2 catchments (Fig. 1b of MS). Assuming that the stream nitrate eluted from FK1 catchment was stable during the flow path from station A to station B. The concentration of stream nitrate eluted from the FK2 catchment was determined by applying Eq. (9):

$$[NO_3^-]_{FK2} = ([NO_3^-]_{FK1+FK2} * F_{FK1+FK2} - [NO_3^-]_{FK1} * F_{FK1}) / F_{FK2} \qquad (9)$$

where $F_{FK1}$, $F_{FK2}$, and $F_{FK1+FK2}$ denote the flux of stream water eluted from the FK1, FK2 (only), and FK1+FK2 catchment, respectively. $[NO_3^-]_{FK1}$, $[NO_3^-]_{FK2}$, and

[NO$_3^-$]$_{FK1+FK2}$ denote the concentration of stream nitrate eluted from the FK1, FK2 (only), and FK1+FK2 catchment, respectively. In this study, the flow rates measured at stations A and B on 2021/01/15 by using the salt dilution method (Sappa et al., 2015) was used for F$_{FK1}$ (0.85 L/s) and F$_{FK1+FK2}$ (4.75 L/s), respectively, and the measured [NO$_3^-$] at stations A and B was used for [NO$_3^-$]$_{FK1}$ and [NO$_3^-$]$_{FK1+FK2}$, respectively. Because the relation between the measured flow rates was comparable with the relation between the catchment area of FK1 (14 ha) and that of FK1+FK2 (76 ha), we concluded that the measured flow rates of 0.85 L/s and 4.75 L/s were reasonable as for those representing the F$_{FK1}$ and F$_{FK1+FK2}$, respectively. According to the mass balance of water, we can estimate the F$_{FK2}$ eluted from the FK2 catchment only to be 3.90 L/s.

Assuming that the stream nitrate eluted from FK1 catchment was stable during the flow path from station A to station B, the $\delta^{15}$N, $\delta^{18}$O, and $\Delta^{17}$O values of stream nitrate eluted from the FK2 catchment only were determined by applying Eq. (10):

$\delta_{FK2} = (\delta_{FK1+FK2} * $ [NO$_3^-$]$_{FK1+FK2} * F_{FK1+FK2} - \delta_{FK1} * $ [NO$_3^-$]$_{FK1} * F_{FK1}) / ($ [NO$_3^-$]$_{FK2} * F_{FK2})$                     (10)

where $\delta_{FK1}$, $\delta_{FK2}$, and $\delta_{FK1+FK2}$ denote the $\delta^{15}$N (or $\delta^{18}$O or $\Delta^{17}$O) of stream nitrate eluted from the FK1, FK2, and FK1+FK2 catchment, respectively. The $\delta^{15}$N (or $\delta^{18}$O or $\Delta^{17}$O) values of stream nitrate measured at stations A and B were used for $\delta_{FK1}$ and $\delta_{FK1+FK2}$, respectively.

Sappa, G., Ferranti, F. and Pecchia, G. M.: Validation Of Salt Dilution Method For Discharge Measurements In The Upper Valley Of Aniene River (Central Italy), Recent Adv. Environ. Ecosyst. Dev., (October 2015), 42–48, 2015.

**>L126: How were the boundaries between the FK1 and FK2 catchments determined? Fig. 1 indicates that these sites are both located along the same stream in the same catchment.**

Firstly, we determined the sampling point in the map by using the GPS data (33.39.31.2689, 130.32.55.0910 for FK1; 33.39.20.9586, 130.32.18.8808 for FK2) (Fig. 4a). Then, we connected the ridge line and the upstream sampling point, which the area (orange) is the FK1 catchment (Fig. 4b). Lastly, by using the same method, the FK2 catchment area was drawn in Fig 4c.

[Figure]

[Figure]

[Figure]

**Figure 4.** The maps showing how we determined the boundary line of the FK1 and FK2 forested catchments.

January 19, 2023
**Response to the referee #2:**

1. **Reviewer #1 raises a significant point about the fact that high atmospheric N leaching rates can also be caused by hydrology (fast leaching rates) rather than biology (slow production rates). These two scenarios can be thought of as 'kinetic limitation' (not enough time for atmospheric N processing) v 'capacity limitation' (not enough biology to process all received atmospheric N),** *sensu* **(Lovett and Goodale, 2011). These two competing explanations could not be distinguished based solely on correlations with rainfall amounts. This is because transit time of NO3- through the canopy, soils, and vadose zone will depend on multiple factors, which include rainfall amount as well as soil types, vegetation root structures, and antecedent moisture conditions. The site descriptions, data analysis, and discussion need to be expanded to adequately address the kinetic limitation hypothesis for Matm/Datm dynamics.**

It is difficult to explain the high concentration of stream nitrate ($[NO_3^-]$) and the high export flux of nitrate ($M_{total}$) by "kinetic limitation" alone, even though high $M_{atm}/D_{atm}$ ratios can be explained by "kinetic limitation" such as a rapid leaching rate, since the majority of nitrate eluted from the catchments was $NO_3^-_{re}$ that had been produced by microbial nitrification. Alternatively, "capacity limitation" can explain both high $[NO_3^-]$ and high $M_{atm}/D_{atm}$ ratios, simultaneously. Significant correlations ($P < 0.0001$) between $M_{total}$ and $M_{atm}/D_{atm}$ ratios in the eleven catchments supported "capacity limitation" as the leading cause of the high $M_{total}$ in FK1 catchment.

In addition, Chiwa (2020) reported the bulk deposition rate of atmospheric $NO_3^-$ and $NH_4^+$ for recent ten years observation was 4.7 and 5.6 kg ha$^{-1}$ yr$^{-1}$ at FK catchments, respectively, and was 3.4 and 4.3 kg ha$^{-1}$ yr$^{-1}$ at MY catchment, respectively. On the other hand, the export flux of total nitrate ($M_{total}$) from FK1 and MY catchment was 13.8 and 3.3 kg ha$^{-1}$ yr$^{-1}$, respectively. As a result, compared to MY catchment, FK1 catchment was a net source for N, which also suggest that FK1 catchment was 'capacity limitation' rather than 'kinetic limitation'.

Furthermore, the old age of the plantation in the FK1 catchment also supported that the catchment exhibited "capacity limitation" as opposed to "kinetic limitation". We would like to add this discussion to the revised manuscript as follows (P24/L474-L486):

The differences in the residence time of water in each catchment could also impact the $M_{atm}/D_{atm}$ ratio, as the residence time of water in forested catchments ranges from one month to more than one year (Asano et al., 2002; Farrick and Branfireun, 2015; Kabeya et al., 2008; Rodgers et al., 2005; Soulsby et al., 2006; Tetzlaff et al., 2007). It is difficult to explain high $[NO_3^-]$ and high $M_{total}$ eluted from the catchment by the residence time of water alone, while the $M_{atm}/D_{atm}$ ratio could be higher in catchments

with a shorter water residence time, as the majority of nitrate eluted from the catchment with a high $M_{atm}/D_{atm}$ ratio was $NO_3^-{}_{re}$ produced by microbial nitrification. The significant correlation between $M_{total}$ and $M_{atm}/D_{atm}$ ratios ($P < 0.0001$; Fig. 3a) supported nitrogen saturation as the leading cause of high $M_{total}$ in catchments with a high $M_{atm}/D_{atm}$ ratio. Additionally, the high loading of atmospheric nitrogen, the type of plantation, and the old age of plantation in the FK1 catchment all supported the conclusion that the FK1 catchment was under the nitrogen saturation.]

2. **Based on Fig. 1 supplied in the response to reviewer comments there is a strong inverse relationship between gross nitrification rate and Matm/Datm (i.e., more nitrification means lower export of atmospheric N). It is only the inclusion of literature values that breaks down the relationship. So why is this? A robust discussion that addresses how (or how not) the high gross nitrification rates fit, or don't, the interpretation that Matm/Datm represents differences in catchment N saturation status.**

[Figure]

The figure was derived in response to a request from Reviewer #1, who concerned the obseved low demand on atmospheric nitrate, thus the high $M_{atm}/D_{atm}$ ratio could be caused by high gross nitrification rate (GNR) in the catchments, FK1 in particular. We have discussed the hypothesis in our reply to referee 1.

First of all, the GNR and $M_{atm}/D_{atm}$ ratios exhibited an inverse correlation instead of a positive correlation, which indicates that the hypothesis was not supported. In addition, the GNR in each catchment estimated from the $\Delta^{17}O$ of stream nitrate eluted from each catchment was generally inaccurate, as explained below. (Ding et al., 2023).

The GNR had been estimated by applying Eq. (1) (Riha et al., 2014; Fang et al., 2015; Hattori et al., 2019; Huang et al., 2020):

$$GNR = D_{atm} \times (\Delta^{17}O(NO_3^-)_{atm} - \Delta^{17}O(NO_3^-)_{stream}) / \Delta^{17}O(NO_3^-)_{stream} \qquad (1)$$

where $D_{atm}$ denote the deposition flux of atmospheric nitrate ($NO_3^-{}_{atm}$) into the catchments, $\Delta^{17}O(NO_3^-)_{atm}$ and $\Delta^{17}O(NO_3^-)_{stream}$ denote the $\Delta^{17}O$ value of $NO_3^-{}_{atm}$ and

stream nitrate, respectively.

To obtain Eq. (1), $\Delta^{17}O(NO_3^-)_{stream}$ must be equal to $\Delta^{17}O$ of $NO_3^-$ consumed in each catchment. The actual $\Delta^{17}O$ of $NO_3^-$ consumed in each catchment (soil $NO_3^-$), however, is always higher than $\Delta^{17}O(NO_3^-)_{stream}$ in forested catchments (Hattori et al., 2019), so Eq. (1) always overestimates GNR (Ding et al., 2023). Almost all $NO_3^-{}_{atm}$ deposited onto MY catchment was consumed within the catchment contrary to the FK1 and FK2 catchments. As a result, the differences between $\Delta^{17}O(NO_3^-)_{stream}$ and $\Delta^{17}O$ of $NO_3^-$ consumed in MY catchment should be larger than those in FK1 and FK2 catchments. Thus, Eq. (1) particularly overestimated GNR in the MY catchment.

Ding, W., Tsunogai, U., and Nakagawa, F.: Ideas and perspectives: Errors associated with the gross nitrification rates in forested catchments calculated from the triple oxygen isotopic composition ($\Delta^{17}O$) of stream nitrate, Biogeosciences Discuss. [preprint], https://doi.org/10.5194/bg-2022-236, in review, 2023.

3. **More details are needed in the methods section about how uncertainties were incorporated into the findings. The Matm/Datm calculations rely on several assumptions that needed to be made in order to account of lack of data (streamflow) or overlapping measurement periods (atmospheric sampling did not occur on the same years as stream water sampling). There are accordingly a number of significant sources of uncertainty incorporated into the Matm/Datm calculations: the relationship between precip amount and streamflow (which itself incorporates a number of uncertainties: the relationship between temperature and evapotranspiration, potential rate of loss to groundwater), the interannual consistency of 17O of atmospheric nitrate, and the spatial consistency in the amount of rainfall and the 17O of atmospheric nitrate. It is therefore essential to critically evaluate the potential magnitude of impact these assumptions have on the resultant Matm/Datm values. A sensitivity analysis needs to be performed for each parameter, and these ranges need to be clearly represented in the figures, tables, and text.**

We would like to include an appendix detailing the calculation of these uncertainties, as shown below (P19/L365-L367; P26-P28/L525-L555):

Appendix A: Calculating of uncertainties in the values of $[NO_3^-{}_{atm}]$, $M_{atm}$, and $M_{atm}/D_{atm}$ ratio

The uncertainty in the values of $[NO_3^-{}_{atm}]$ was estimated from the uncertainties in the $\Delta^{17}O$ values of stream nitrate ($\Delta^{17}O$) and $NO_3^-{}_{atm}$ ($\Delta^{17}O_{atm}$) according to the divisive equation of error propagation (A1):

$$\sigma_{[NO_3^-{}_{atm}]}=[NO_3^-] * \sqrt{(\frac{1}{\Delta^{17}O_{atm}}*\sigma_{\Delta^{17}O})^2 + (\frac{\Delta^{17}O}{\Delta^{17}O_{atm}{}^2}*\sigma_{\Delta^{17}O_{atm}})^2} \tag{A1}$$

where $\sigma_{[NO_3^-{}_{atm}]}$, $\sigma_{\Delta^{17}O}$, and $\sigma_{\Delta^{17}O_{atm}}$ denote the uncertainties in $[NO_3^-{}_{atm}]$, $\Delta^{17}O$ values of stream nitrate, and $\Delta^{17}O$ values of $NO_3^-{}_{atm}$, respectively. The standard error of the mean (SE) of ±0.1 ‰ and the areal/seasonal variations of ±3 ‰ was used in calculating $\sigma_{\Delta^{17}O}$ and $\sigma_{\Delta^{17}O_{atm}}$, respectively. As a result, the uncertainty in $[NO_3^-{}_{atm}]$ ($\sigma_{[NO_3^-{}_{atm}]}$) was ±1.30, ±0.67, and ±0.03 μM at FK1, FK2, and MY catchments, respectively.

The uncertainty in the values of $M_{atm}$ was estimated from the uncertainties in $[NO_3^-{}_{atm}]$ and in $F_{stream}$ according to the multiplicative equation of error propagation (A2):

$$\sigma_{M_{atm}}=\sqrt{(F_{stream}*\sigma_{[NO_3^-{}_{atm}]})^2 + ([NO_3^-{}_{atm}]*\sigma_{F_{stream}})^2} \tag{A2}$$

where $\sigma_{M_{atm}}$, $\sigma_{[NO_3^-{}_{atm}]}$, and $\sigma_{F_{stream}}$ denote the uncertainties in $M_{atm}$, $[NO_3^-{}_{atm}]$, and $F_{stream}$, respectively. Komatsu et al. (2008) proposed the uncertainty in $F_{stream}$ to be ±162.3 mm when using the water balance method in estimating $F_{stream}$. Here, the uncertainty in $M_{atm}$ ($\sigma_{M_{atm}}$) was ±2.1, ±1.0, and ±0.1 mmol m$^{-2}$ yr$^{-1}$ at FK1, FK2, and MY catchments, respectively.

The uncertainty in $M_{atm}/D_{atm}$ ratio was estimated from the uncertainties in $M_{atm}$ and in $D_{atm}$ according to the divisive equation of error propagation (A3):

$$\sigma_{M_{atm}/D_{atm} \text{ ratio}}=\sqrt{(\frac{1}{D_{atm}}*\sigma_{M_{atm}})^2 + (\frac{M_{atm}}{D_{atm}{}^2}*\sigma_{D_{atm}})^2} \tag{A3}$$

where $\sigma_{M_{atm}/D_{atm} \text{ ratio}}$, $\sigma_{M_{atm}}$, and $\sigma_{D_{atm}}$ denote the uncertainty in $M_{atm}/D_{atm}$ ratio, $M_{atm}$, and $D_{atm}$, respectively. Comparing the deposition rate of $NO_3^-{}_{atm}$ obtained at the other atmospheric monitoring stations nearby, the uncertainty of 20 % was adopted for those of $D_{atm}$ in each catchment, which corresponds to the uncertainty in $D_{atm}$ of ±13.9, ±13.9, ±8.0 mmol m$^{-2}$ yr$^{-1}$ at FK1, FK2, and MY catchments, respectively. As a result, the uncertainty in $M_{atm}/D_{atm}$ ratio was ±4.1 %, ±2.0 %, and ±0.4 % at FK1, FK2, and MY catchments, respectively.

These uncertainties were shown in the figures, tables, and text in the revised manuscript.

4. **I am still worried about the reliance on, essentially, rainfall and average annual catchment temperature to calculate downstream NO3- discharge. The relationship between rainfall amounts and stream discharge is generally**

**highly complex, and affected by a number of factors such as catchment slope, soils, vegetation, and groundwater connectivity. These factors need to be robustly and quantitatively addressed (i.e., a hydrodynamic model is needed) given how important Fstream is to Matm, and thus the interpretation of systems as N saturated.**

First of all, the variation in the stream water flux ($F_{stream}$) has small effect on the calculation of $M_{atm}$ and $M_{total}$ as compared to the variations of $[NO_3^-{}_{atm}]$ and $[NO_3]$ in monsoon regions with high precipitation, where the majority of rainwater elutes as stream water. Komatsu et al. (2008) compiled the precipitation, $F_{stream}$, and evapotranspiration (E) determined in 43 forested catchments in Japan (Fig. 1). The evapotranspiration (E = precipitation − $F_{stream}$) in the 43 forested catchments ranged from 109 to 1267 mm, with an average E of 733 mm and standard deviation (SD) of 218 mm, which corresponds to a 30% coefficient of variation (CV). In contrast, the CVs of $[NO_3^-{}_{atm}]$ and $[NO_3^-]$ compiled for this study were 99% and 92%, respectively. Consequently, $[NO_3^-{}_{atm}]$ and $[NO_3^-]$ in the stream water, and not $F_{stream}$, are the primary determinants of $M_{atm}$ and $M_{total}$.

In addition, the water balance method in forested catchments has been well-established in previous research (e.g., Komatsu et al., 2008; Zhang et al., 2001; Harder et al., 2007; Combalicer et al., 2008; Milly, 1994), and the method has been used in quantifying the flux of stream water ($F_{stream}$) and evapotranspiration flux of water in numerous past studies (e.g., Wang et al., 2022; Che et al., 2022; Clark et al., 2014; Kozii et al., 2020). Komatsu et al. (2008) confirmed that the estimated $F_{stream}$ derived from the water balance method is consistent with the $F_{stream}$ observed in three forested catchments (Fig. 2). As a result, we employed the water balance method proposed by Komatsu et al. (2008) in quantifying the $F_{stream}$ in the catchments. We would like to add the following information to the revised manuscript (P14/L259-L262):

They also confirmed that the estimated $F_{stream}$ using the model corresponded well with the observed $F_{stream}$ in three forested catchments, with the estimated errors of less than 6 %. As a result, we utilized the water balance method proposed by Komatsu et al. (2008) to quantify the $F_{stream}$ in each catchment.

Additionally, Komatsu et al. (2008) proposed that the standard error when employing the method to estimate $F_{stream}$ was 162.3 mm, which was factored into the uncertainty of $M_{total}$, $M_{atm}$, and $M_{atm}/D_{atm}$ ratio in this study.

[Figure]

**Figure 1.** Locations of the 43 forest catchments compiled by Komatsu et al. (2008).

[Figure]

**Figure 2.** Comparisons between observed $F_{stream}$ and estimated $F_{stream}$ by applying the water balance method proposed by Komatsu et al. (2008) in three different forested catchments in Japan. Precipitation data P is also shown (Komatsu et al., 2008).

5. **As a consequence of the above (big) assumption that Fstream = precipitation – evapotransiration, the Matm/Datm ratio is essentially: $([NO3]stream*(P-E))/([NO3]bulk*P)$ (ignoring for a moment the calculations around dry and gaseous deposition). This really is then a almost directly a comparison of the concentration of 17O-NO3- measured in stream water over a few years relative to the concentrations of 17O-NO3- measured in the rain over the previous decade, with correction factor for the average annual temperature of the catchment (used to calculate E). Without a more robust approach to uncertainty and stream flow, and a more nuanced discussion of these uncertainties, it is hard to draw any conclusions about ecosystem N saturation from these values. It is also difficult to justify statistical analyses comparing temperature, precipitation, and discharge to Matm/Datm, given that all three parameters are directly used to calculate the ratio (and indeed that temperature and precipitation are themselves used to calculate discharge).**

The water balance method is well established in Japan, as was stated previously. In addition, the uncertainties associated with the estimated $M_{atm}/D_{atm}$ ratios included all parameter-related uncertainties.

Significantly elevated $[NO_3^-{}_{atm}]$ and a high $M_{atm}/D_{atm}$ ratio were found in stream water eluted from the FK1 catchment with significantly elevated $[NO_3^-]$ in this study. This discovery is without a doubt significant in elucidating the causes of the high $[NO_3^-]$ in the forested stream.

6. **I am still very confused about the relationship between FK1 and FK2. Are these in the same catchment or different catchments?**

Thank you for your questions. They are different catchments. Therefore, we have revised the manuscript to clarify this.

**Does one flow into the other (referred to as upstream v downstream sites at some points), or do they flow off different sides of a ridge?**

One flows into the other. We would like to add the stream flow direction to the revised map. The blue arrows indicate the flow direction of stream water.

[Figure]

If the latter, does this affect the amount of precipitation received at both sites? **If the former, should these really be considered as independent sites?**

Because there are significant differences between concentrations, $\delta^{18}O$ and $\Delta^{17}O$ of the stream nitrate in catchments FK1 and FK2 (all $P < 0.001$). Here, catchments FK1 and FK2 should be considered independent catchments.

**It also seems the reliance on rain and temperature to determine flow would have a big impact here.**

Because the central distance between FK1 and FK2 catchment was no more than 2 km, the differences in rain and temperature between FK1 and FK2 catchment can be ignored.

**Are the streams actually the same size, as would presumably be determined by these calculations?**

The flow rates measured at stations A and B on 2021/01/15 was 0.85 L/s (flow rate of FK1) and 4.75 L/s (flow rate of FK1+FK2), respectively. As a result, the stream flow rate of FK1 catchment was 0.85 L/s, and the stream flow rates of FK2 catchment can be calculated as 3.90 L/s, respectively. Because the relation between the measured flow rates was comparable with the relation between the catchment area of FK1 (14 ha) and that of FK2 (62 ha), we concluded that the measured flow rates on 2021/01/15 were reasonable. We have discussed this point in section 2.7 of the manuscript (P15-P16/L285-L313).

We would like to thank you for the helpful comments and suggestions. We hope that our responses to your comments and questions are satisfactory.

Sincerely,
Weitian Ding

Ph.D. student
Graduate School of Environmental Studies,
Nagoya University
Furo-cho, Chikusa-ku, Nagoya,
464-8601, JAPAN
Phone: +81-70-4436-3157
E-mail: ding.weitian.v2@s.mail.nagoya-u.ac.jp

**Reference**

Che, X., Jiao, L., Qin, H., and Wu, J.: Impacts of Climate and Land Use/Cover Change on Water Yield Services in the Upper Yellow River Basin in Maqu County, Sustainability, 14, 10363, https://doi.org/10.3390/su141610363, 2022.

Clark, K. E., Torres, M. A., West, A. J., Hilton, R. G., New, M., Horwath, A. B., Fisher, J. B., Rapp, J. M., Robles Caceres, A., and Malhi, Y.: The hydrological regime of a forested tropical Andean catchment, Hydrology and Earth System Sciences, 18, 5377–5397, https://doi.org/10.5194/hess-18-5377-2014, 2014.

Combalicer, E. A., Lee, S. H., Ahn, S., Kim, D. Y., and Im, S.: Modeling water balance for the small-forested watershed in Korea, KSCE J Civ Eng, 12, 339–348, https://doi.org/10.1007/s12205-008-0339-y, 2008.

Ding, W., Tsunogai, U., and Nakagawa, F.: Ideas and perspectives: Errors associated with the gross nitrification rates in forested catchments calculated from the triple oxygen isotopic composition ($\Delta^{17}O$) of stream nitrate, Biogeosciences Discuss. [preprint], https://doi.org/10.5194/bg-2022-236, in review, 2023.

Fang, Y., Koba, K., Makabe, A., Takahashi, C., Zhu, W., Hayashi, T., Hokari, A. A., Urakawa, R., Bai, E., Houlton, B. Z., Xi, D., Zhang, S., Matsushita, K., Tu, Y., Liu, D., Zhu, F., Wang, Z., Zhou, G., Chen, D., Makita, T., Toda, H., Liu, X., Chen, Q., Zhang, D., Li, Y. and Yoh, M.: Microbial denitrification dominates nitrate losses from forest ecosystems, Proc. Natl. Acad. Sci. U. S. A., 112(5), 1470–1474, doi:10.1073/pnas.1416776112, 2015.

Harder, S. V., Amatya, D. M., Callahan, T. J., Trettin, C. C., and Hakkila, J.: Hydrology and Water Budget for a Forested Atlantic Coastal Plain Watershed, South Carolina1, JAWRA Journal of the American Water Resources Association, 43, 563–575, https://doi.org/10.1111/j.1752-1688.2007.00035.x, 2007.

Hattori, S., Nuñez Palma, Y., Itoh, Y., Kawasaki, M., Fujihara, Y., Takase, K. and Yoshida, N.: Isotopic evidence for seasonality of microbial internal nitrogen cycles in a temperate forested catchment with heavy snowfall, Sci. Total Environ., 690, 290–299, doi:10.1016/j.scitotenv.2019.06.507, 2019.

Huang, S., Wang, F., Elliott, E. M., Zhu, F., Zhu, W., Koba, K., Yu, Z., Hobbie, E. A., Michalski, G., Kang, R., Wang, A., Zhu, J., Fu, S. and Fang, Y.: Multiyear Measurements on $\Delta^{17}O$ of Stream Nitrate Indicate High Nitrate Production in a Temperate Forest, Environ. Sci. Technol., 54(7), 4231–4239, doi:10.1021/acs.est.9b07839, 2020.

Komatsu, H., Maita, E., and Otsuki, K.: A model to estimate annual forest evapotranspiration in Japan from mean annual temperature, 330–340, https://doi.org/10.1016/j.jhydrol.2007.10.006, 2008.

Kozii, N., Haahti, K., Tor-ngern, P., Chi, J., Hasselquist, E. M., Laudon, H., Launiainen, S., Oren, R., Peichl, M., Wallerman, J., and Hasselquist, N. J.: Partitioning growing season water balance within a forested boreal catchment using sap flux, eddy covariance, and a process-based model, Hydrology and Earth System Sciences, 24, 2999–3014, https://doi.org/10.5194/hess-24-2999-2020, 2020.

Milly, P. C. D.: Climate, soil water storage, and the average annual water balance, Water Resources Research, 30, 2143–2156, https://doi.org/10.1029/94WR00586, 1994.

Nakagawa, F., Tsunogai, U., Obata, Y., Ando, K., Yamashita, N., Saito, T., Uchiyama, S., Morohashi, M. and Sase, H.: Export flux of unprocessed atmospheric nitrate from temperate forested catchments: A possible new index for nitrogen saturation, Biogeosciences, 15(22), 7025–7042, doi:10.5194/bg-15-7025-2018, 2018.

Riha, K. M., Michalski, G., Gallo, E. L., Lohse, K. A., Brooks, P. D. and Meixner, T.: High Atmospheric Nitrate Inputs and Nitrogen Turnover in Semi-arid Urban Catchments, Ecosystems, 17(8), 1309–1325, doi:10.1007/s10021-014-9797-x, 2014.

Wang, Y., Wang, H., Liu, G., Zhang, J., and Fang, Z.: Factors driving water yield ecosystem services in the Yellow River Economic Belt, China: Spatial heterogeneity and spatial spillover perspectives, Journal of Environmental Management, 317, 115477, https://doi.org/10.1016/j.jenvman.2022.115477, 2022.

Zhang, L., Dawes, W. R., and Walker, G. R.: Response of mean annual evapotranspiration to vegetation changes at catchment scale, Water Resources Research, 37, 701–708, https://doi.org/10.1029/2000WR900325, 2001.